

# Fractional magnetization plateaux of a spin-1/2 Heisenberg model on the Shastry-Sutherland lattice: Effect of quantum XY interdimer coupling

**Taras Verkholyak**[1][⋆] **and Jozef Strečka**[2]

**1** Institute for Condensed Matter Physics, National Academy of Sciences of Ukraine,
1 Svientsitskii Street, Lviv-11, 79011, Ukraine
**2** Department of Theoretical Physics and Astrophysics, Faculty of Science,
P. J. Šafárik University, Park Angelinum 9, 040 01, Košice, Slovakia

⋆ werch@icmp.lviv.ua

## Abstract

Spin-1/2 Heisenberg model on the Shastry-Sutherland lattice is considered within the many-body perturbation theory developed from the exactly solved spin-1/2 Ising-Heisenberg model with the Heisenberg intradimer and Ising interdimer interactions. The former model is widely used for a description of magnetic properties of the layered compound $SrCu_2(BO_3)_2$, which exhibits a series of fractional magnetization plateaux at sufficiently low temperatures. Using the novel type of many-body perturbation theory we have found the effective model of interacting triplet excitations with the extended hard-core repulsion, which accurately recovers 1/8, 1/6 and 1/4 magnetization plateaux for moderate values of the interdimer coupling. A possible existence of a striking quantum phase of bound triplons is also revealed at low enough magnetic fields.

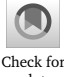

# 1 Introduction

The spin-1/2 quantum Heisenberg model on the Shastry-Sutherland lattice [1], which is traditionally referred to as the Shastry-Sutherland model, is the modern playground for the study of the complex structure of quantum matter emergent in low-dimensional spin systems. Its structure is characterized by the two-dimensional array of mutually orthogonal spin dimers [2]. The competition between the antiferromagnetic intradimer $J$ and interdimer $J'$ couplings brings on the short-range correlated singlet-dimer and singlet-plaquette phases in the ground state [1,3]. The application of the magnetic field invokes several subtle fractional magnetization plateaux with complex ordering of localized triplon excitations (see Ref. [4] for a review). The theoretical study of the Shastry-Sutherland model generally represents a complex problem tackled by many numerical methods like CORE [5], pCUTs [6–8], tensor network iPEPS [9–11], and others (for a review see Ref. [2]). Besides, the possible emergence of bound states [9,12–17], topological triplon modes [18–20] and Bose-Einstein condensation [2,21] is of the current research interest. Recently, the numerical variational approach shed light on the quantum phases emerging at the boundary of the singlet-dimer phase [22]. The study of thermodynamics is even more complex due to a problem with thermal averaging. Most recently the quantum Monte Carlo (QMC) method has been specifically adapted to the Shastry-Sutherland model in a dimer basis in order to avoid a sign problem within QMC simulations of this frustrated quantum spin system [23]. The dimer basis has been also utilized to calculate the thermodynamic properties by the suggested numerical methods based on typical pure quantum states and infinite projected entangled-pair states [17]. Temperature-driven phase transitions and crossovers are being another challenging task, which nowadays attract considerable attention [24–26].

The most prominent experimental representative of the Shastry-Sutherland model is the layered magnetic material $SrCu_2(BO_3)_2$. Being rediscovered by Kageyama *et al.* [27], this magnetic compound provides a long sought after experimental realization of the singlet-dimer phase at zero magnetic field and several exotic quantum phases, which are manifested in a low-temperature magnetization curve as a series of fractional magnetization plateaux [10,28,29]. What is even more, the ratio between the intradimer and interdimer couplings in this magnetic compound is quite close to a phase boundary between the singlet-dimer phase and the singlet-plaquette phase [3]. It turns out that the external pressure indeed paves the way for tuning the relative ratio between the interdimer and intradimer coupling constants through the crystal deformation and hence, one may observe at low enough temperatures a phase transition between the singlet-dimer, singlet-plaquette and Néel phases [30]. Physical manifestations of these phases in $SrCu_2(BO_3)_2$ is under intense current experimental and theoretical research [31–35]. Nevertheless, even the magnetic behavior at low magnetic fields is still not completely understood. In Ref. [28] the fractional magnetization plateaux at 1/8, 2/15, 1/6, 1/4 of the saturation magnetization were revealed by nuclear magnetic resonance and torque measurements at $T = 60$ mK, while in Ref. [10] magnetization measurements at ultrahigh magnetic fields gave evidence for the 1/8, 1/4, 1/3, 1/2 magnetization plateaux recorded at $T = 2.4$ K. It is noteworthy that the Dzyaloshinskii-Moriya interaction is also important for the proper description of the magnetic properties of $SrCu_2(BO_3)_2$ [18,21,36–38]. In particular, it

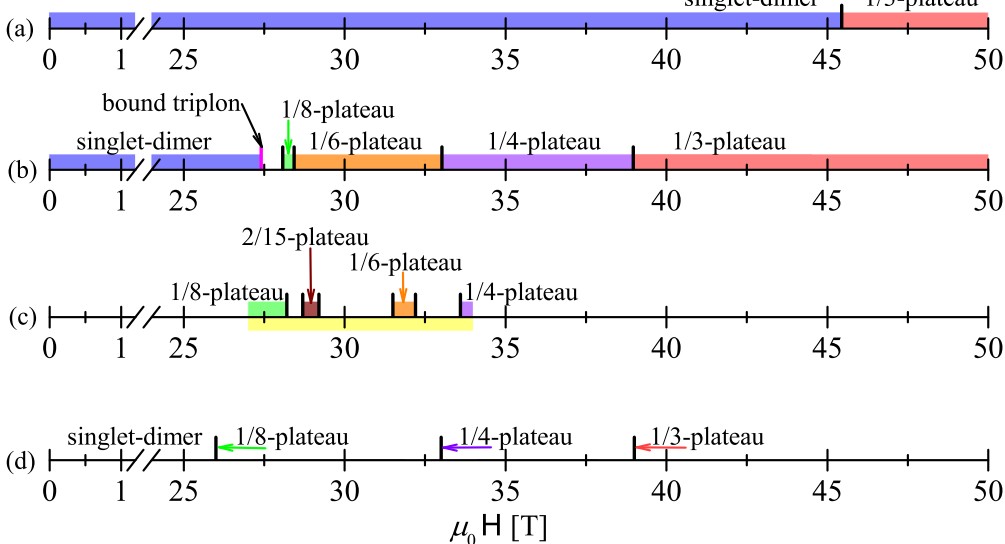

Figure 1: A schematic phase diagram for the Shastry-Sutherland model (panels a, b) and for the related magnetic compound $SrCu_2(BO_3)_2$ (panels c, d). Different color strips indicate different plateaux. (a) The Ising-Heisenberg model on the Shastry-Sutherland lattice with the intradimer $J/k_B = 85.4$ K and interdimer $J'/k_B = 54.1$ K couplings, the thick vertical bar corresponds to $\mu_0 H_{c1} = 45.5$ T, recovered from Eq. (4) using the relation $\mu_0 H = \mu_0 h/g\mu_B$; (b) the results of the perturbative approach given in Secs. 3 and 4 for the same interactions as specified in above: the vertical bars denote the transition fields between different plateau phases of localized triplons from left to right $\mu_0 H_{0-1/8} = 28.1$ T, $\mu_0 H_{1/8-1/6} = 28.4$ T, $\mu_0 H_{1/6-1/4} = 33$ T, $\mu_0 H_{1/4-1/3} = 39$ T recovered from Eq. (8). A magenta bar denotes the critical field for the bound-triplon phase $\mu_0 H_{bound-t} = 27.4$ T (see Sec. 4 and Eq. (12) therein); (c) the result of the NMR measurements in the field range $27-34$ T [28]: 1/8 plateau is detected below 28.2 T, 2/15, 1/6-plateaux are found within the field ranges $28.7-29.2$ T and $31.5-32.2$ T, respectively, while 1/4 plateau arises at 33.6 T; (d) the results of the magnetization measurements reported in Ref. [10]: 1/8, 1/4 and 1/3 plateaux start at 26 T, 33 T and 39 T indicated by arrows pointing to vertical bars.

leads to the non-trivial topological band structure of triplons [18].

It should be noted that $SrCu_2(BO_3)_2$ is not the only example of the physical realization of the Shastry-Sutherland model. The magnetic structure of $(CuCl)Ca_2Nb_3O_{10}$ is also consistent with the Shastry-Sutherland model, but the ferromagnetic character of the interdimer coupling regrettably prevents the emergence of fractional plateaux in the respective low-temperature magnetization curves [39,40]. On the other hand, the rare-earth tetraborides $RB_4$ ($R$ = Dy, Er, Tm, Tb, Ho) afford another intriguing class of the magnetic materials, which display a complex structure of fractional magnetization plateaux inherent to the anisotropic Heisenberg model on the Shastry-Sutherland lattice with a rather strong Ising-type anisotropy [41–47]. While the antiferromagnetic spin-1/2 Ising model on the Shastry-Sutherland lattice is capable of reproducing the fractional 1/3-plateau only [48], the possible exchange coupling between localized Ising spins and the spins of conducting electrons in the metallic rare-earth tetraborides may be essential for a description of their magnetic properties [49,50].

In the present work we consider the Shastry-Sutherland model by means of the strong-coupling approach based on the perturbative treatment of $XY$ interdimer interaction. The main result of this many-body perturbation calculation is summarized in Fig. 1. The Ising-

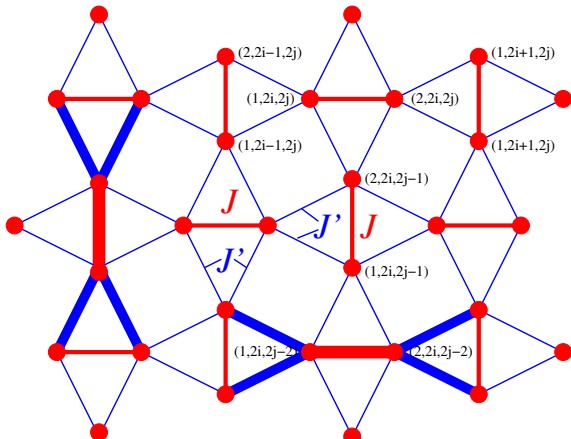

Figure 2: A schematic representation of the Shastry-Sutherland model defined through the Hamiltonian (1). Thick (red) lines denote the stronger intradimer coupling $J$, while thin (blue) lines show the interdimer coupling $J'$. Bold lines mark clusters of three subsequent strongly correlated dimers in the Ising-Heisenberg model given by the Hamiltonian (2).

Heisenberg model on the Shastry-Sutherland lattice shows only the 1/3-plateau above the zero magnetized singlet-dimer phase, see Fig. 1(a). The perturbative treatment of the $XY$ part of the interdimer coupling is capable of reproducing the series of the fractional 1/8-, 1/6-, and 1/4-plateaux presented in Fig. 1(b), which shows a rather good agreement with the experimental measurements summarized in panels (c) and (d) according to Refs. [10, 28]. In Sec. 2 we define the model and expose the applied perturbative method. In Sec. 3 we discuss in particular the results of the second-order perturbation theory with the main emphasis laid on the phases of localized triplons emergent in the ground-state phase diagram. We also demonstrate to what extent the sequence of fractional plateaux in $SrCu_2(BO_3)_2$ can be described. The quantum correlated phases are considered in Sec. 4, while the most interesting findings are summarized in Sec. 5. Some specific technical details of the calculation procedure are shown in Appendices.

## 2 Shastry-Sutherland model

Let us consider the spin-1/2 Heisenberg model on the Shastry-Sutherland lattice [1] in an external magnetic field defined by the following Hamiltonian:

$$H = J \sum_{\langle l, m \rangle} \mathbf{s}_l \cdot \mathbf{s}_m + J' \sum_{\langle\langle l, m \rangle\rangle} \mathbf{s}_l \cdot \mathbf{s}_m - h \sum_l s_l^z, \tag{1}$$

where $\langle l, m \rangle$ and $\langle\langle l, m \rangle\rangle$ denote the summation over all intradimer $J$ and interdimer $J'$ interactions, the general site index $l = (n, i, j)$ involves a reference number $n = 1$ or $2$ of the spin in a dimer in addition to two reference numbers $i$ and $j$ specifying the dimer's position in a column and a row of the Shastry-Sutherland lattice, respectively (see Fig. 2 for enumeration of sites). Last, the parameter $h = g\mu_B H$ denotes the standard Zeeman's term, $\mu_B$ is the Bohr magneton, $g$ is the gyromagnetic factor of magnetic ions and $H$ is an external magnetic field.

A usual procedure for the perturbative treatment starts from the limit of non-interacting dimers leaving the weaker interdimer coupling as a perturbation [14, 51]. However, such an approach is slowly converging and requires the higher-order expansion terms. For instance, the expansion up to the third order is able to reproduce only the 1/2 and 1/3 plateaux of the

Shastry-Sutherland model [14]. In the present work we develop the unconventional perturbation theory from the exact solution for the ground state of the spin-1/2 Ising-Heisenberg model on the Shastry-Sutherland lattice with the Heisenberg intradimer and Ising interdimer couplings given by the Hamiltonian [52]:

$$H_{IH} = J \sum_{\langle l,m \rangle} \mathbf{s}_l \cdot \mathbf{s}_m + J' \sum_{\langle\langle l,m \rangle\rangle} s_l^z s_m^z - h \sum_l s_l^z. \tag{2}$$

If the intradimer interaction is assumed to be much stronger than the interdimer interaction, i.e. $J > J'$, it is convenient to utilize the dimer-state basis for a pair of spins coupled by the stronger intradimer interaction:

$$
\begin{aligned}
|0\rangle_{i,j} &= \frac{1}{\sqrt{2}}(|\uparrow\rangle_{1,i,j}|\downarrow\rangle_{2,i,j} - |\downarrow\rangle_{1,i,j}|\uparrow\rangle_{2,i,j}), \\
|1\rangle_{i,j} &= |\uparrow\rangle_{1,i,j}|\uparrow\rangle_{2,i,j}, \\
|2\rangle_{i,j} &= \frac{1}{\sqrt{2}}(|\uparrow\rangle_{1,i,j}|\downarrow\rangle_{2,i,j} + |\downarrow\rangle_{1,i,j}|\uparrow\rangle_{2,i,j}), \\
|3\rangle_{i,j} &= |\downarrow\rangle_{1,i,j}|\downarrow\rangle_{2,i,j}, 
\end{aligned}
\tag{3}
$$

where $|0\rangle_{i,j}$ denotes the singlet-dimer state, and $|1\rangle_{i,j}$, $|2\rangle_{i,j}$, $|3\rangle_{i,j}$ correspond to the triplet-dimer states with the following values of $z$-component of the total spin $S_{i,j}^z = s_{1,i,j}^z + s_{2,i,j}^z = 1, 0, -1$, respectively. Such a representation provides a transparent description of all ground states, which emerge in the spin-1/2 Ising-Heisenberg model on the Shastry-Sutherland lattice given by the Hamiltonian (2) [52]. In fact, the $z$-component of the total spin on a dimer $S_{i,j}^z$ is a well defined quantum number and the Hamiltonian $H_{IH}$ can be brought into a diagonal form by the unitary transformation (see Ref. [52] and Appendix A for further details). Thus, the problem of finding its ground state is turned into the minimization of the diagonalized Hamiltonian with respect to the quantum states on local dimers. It has been verified [52] that the ground state of the spin-1/2 Ising-Heisenberg model on the Shastry-Sutherland lattice (2) for $J' \leq J$ can be characterized by the following four phases: the singlet-dimer phase for $h < h_{c1}$, the stripe 1/3-plateau phase (see Fig. 3) for $h_{c1} < h < h_{c2}$, the checkerboard 1/2-plateau phase for $h_{c2} < h < h_{c3}$ and the saturated paramagnetic phase for $h > h_{c3}$. The critical fields determining the relevant ground-state phase boundaries are explicitly given as follows:

$$
\begin{aligned}
h_{c1} &= 2J - \sqrt{J^2 + J'^2}, \\
h_{c2} &= -J + 2\sqrt{J^2 + J'^2}, \\
h_{c3} &= J + 2J'.
\end{aligned}
\tag{4}
$$

It should be pointed out that the ground state of the spin-1/2 Ising-Heisenberg model on the Shastry-Sutherland lattice is macroscopically degenerate at the first critical field $h_{c1}$ and moreover, this highly degenerate ground-state manifold can be treated as a gas of $S^z = 1$ triplet excitations referred to as triplons [53] emergent on a background of the crystal of singlet dimers. It has been shown [52] that the ground states of the spin-1/2 Ising-Heisenberg model on the Shastry-Sutherland lattice can be constructed from a six-spin cluster composed of three consecutive dimers (see Fig. 2) under the following restriction: no more than one triplon excitation can be located on each six-spin cluster. This hard-core condition follows from the fact that at the critical field between the singlet-dimer and 1/3 plateau phase the ground state configuration can be constructed from the three-dimer clusters defined by the Hamiltonians (C.2), where each cluster may contain no more than one triplet excitation (see

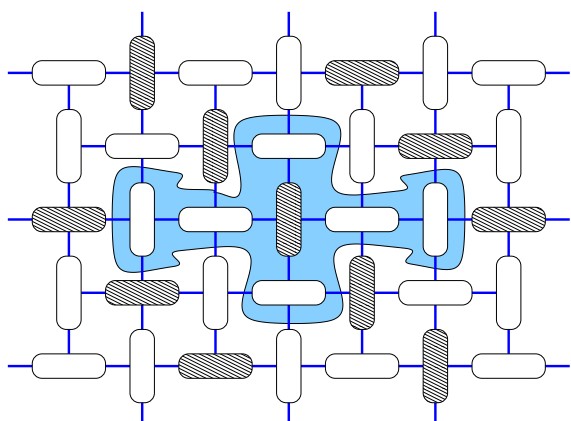

Figure 3: A schematic representation of the stripe 1/3-plateau phase and a hard-core repulsion condition for triplons. Dashed (empty) rounded rectangles denote $S^z = 1$ triplet (singlet) states on dimers. The blue shaded area indicates a hard-core constraint for a central dimer in a triplet state, which excludes triplons on all its four nearest-neighbor dimers as well as two alike oriented further-neighbor dimers along one spatial direction.

Ref. [52] for the details). Consequently, such triplet excitations should obey the hard-core constraint: triplons cannot be created on four adjacent (nearest-neighbor) dimers and such an exclusion rule should be additionally extended for a vertically (horizontally) oriented dimer to its two alike oriented further-neighbor dimers in horizontal (vertical) direction (see Fig. 3).

In the following our attention is focused on the phase boundary between the singlet-dimer and stripe 1/3-plateau phases, i.e. the magnetic-field region sufficiently close to the first critical field $h_{c1}$, where the low-temperature magnetization curve of the spin-1/2 Heisenberg model on the Shastry-Sutherland lattice exhibits the most spectacular features (see Ref. [4] for a review). To this end, the Hamiltonian (2) of the spin-1/2 Ising-Heisenberg model on the Shastry-Sutherland lattice at the critical field $h = h_{c1}$ with exactly known ground state is used as the unperturbed part of the Hamiltonian (1) of the spin-1/2 Heisenberg model on the Shastry-Sutherland lattice. The remaining part of the Hamiltonian (1) involving the $XY$-part of the interdimer interaction is perturbatively treated together with the deviation of the magnetic field from the critical value $h_{c1}$:

$$H' = J'_{xy} \sum_{\langle\langle l,m \rangle\rangle} (s_l^x s_m^x + s_l^y s_m^y) - (h - h_{c1}) \sum_l s_l^z. \tag{5}$$

Here we introduced a separate notation $J'_{xy}$ for the $XY$-part of the interdimer coupling. In all final expressions we set $J'_{xy} = J'$.

## 3 Effective model of triplon excitations

Let us apply the many-body perturbation theory (see for instance Ref. [54] for a general procedure) to the $XY$-part of the interdimer interaction at the phase boundary between the singlet-dimer and stripe 1/3-plateau phase, where the ground state is macroscopically degenerate and can be presented as a lattice gas of triplet excitations on dimers. Within the second-order perturbation expansion one obtains the following effective Hamiltonian when excluding from consideration two triplet states $|2\rangle_{i,j}$, $|3\rangle_{i,j}$ of each dimer while retaining only the singlet state

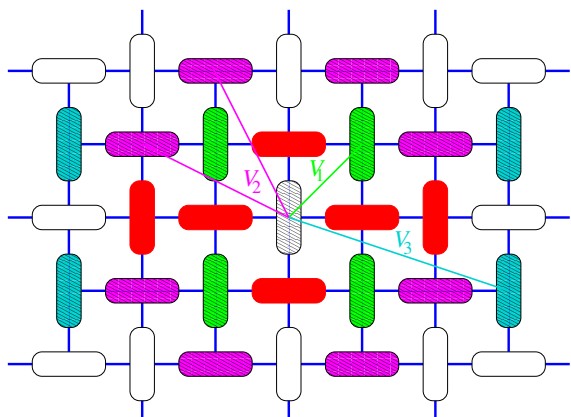

Figure 4: A schematic representation of effective pair interactions between triplons. Red rounded rectangles around the central dimer in the triplet state (dashed rounded rectangle) show the hard-core condition, the colored rectangles indicate the interaction between the central triplon and its further neighbors.

$|0\rangle_{i,j}$ and fully polarized triplet state $|1\rangle_{i,j}$ (see Appendix B for details):

$$H_{eff} = \mathcal{P}_0[E_0 + H_1 + H_t + H_2 + H_3 + \dots]\mathcal{P}_0,$$

$$H_1 = (e_0 + h_{c1} - h)\sum_{\mathbf{l}} n_{\mathbf{l}}, \quad \mathbf{l} = (l_x, l_y),$$

$$H_2 = V_1 \sum_{\langle \mathbf{l},\mathbf{l}'\rangle} n_{\mathbf{l}} n_{\mathbf{l}'} + V_2 \sum_{\langle \mathbf{l},\mathbf{l}'\rangle'} n_{\mathbf{l}} n_{\mathbf{l}'} + V_3 \sum_{\langle \mathbf{l},\mathbf{l}'\rangle''} n_{\mathbf{l}} n_{\mathbf{l}'},$$

$$H_3 = V_{\triangle 1} \sum_{\langle \mathbf{l},\mathbf{l}',\mathbf{l}''\rangle_1} n_{\mathbf{l}} n_{\mathbf{l}'} n_{\mathbf{l}''} + V'_{\triangle 1} \sum_{\langle \mathbf{l},\mathbf{l}',\mathbf{l}''\rangle'_1} n_{\mathbf{l}} n_{\mathbf{l}'} n_{\mathbf{l}''} + V_{\triangle 2} \sum_{\langle \mathbf{l},\mathbf{l}',\mathbf{l}''\rangle_2} n_{\mathbf{l}} n_{\mathbf{l}'} n_{\mathbf{l}''}$$

$$+ V'_{\triangle 2} \sum_{\langle \mathbf{l},\mathbf{l}',\mathbf{l}''\rangle'_2} n_{\mathbf{l}} n_{\mathbf{l}'} n_{\mathbf{l}''} + V''_{\triangle 2} \sum_{\langle \mathbf{l},\mathbf{l}',\mathbf{l}''\rangle''_2} n_{\mathbf{l}} n_{\mathbf{l}'} n_{\mathbf{l}''} + V_{\triangle 3} \sum_{\langle \mathbf{l},\mathbf{l}',\mathbf{l}''\rangle_3} n_{\mathbf{l}} n_{\mathbf{l}'} n_{\mathbf{l}''},$$

$$H_t = t \sum_{i,j=1}^{N}{}' (n_{i,j-1} + n_{i,j+1})(a^+_{i-1,j} a_{i+1,j} + a_{i-1,j} a^+_{i+1,j})$$

$$+ t \sum_{i,j=1}^{N}{}'' (n_{i-1,j} + n_{i+1,j})(a^+_{i,j-1} a_{i,j+1} + a_{i,j-1} a^+_{i,j+1}). \tag{6}$$

The summation symbol $\sum'$ ($\sum''$) is restricted by the constraint $i + j = $ odd ($i + j = $ even) extended over all vertical (horizontal) dimers and $\mathcal{P}_0$ is the projection operator incorporating the hard-core condition for triplons (see Fig. 3 and Eq. (B.4)). Each site of the effective model (6) corresponds to a dimer of the Shastry-Sutherland lattice, and an empty (filled) site $n_{\mathbf{l}} = 0$ ($n_{\mathbf{l}} = 1$) of the effective model is assigned to the singlet state $|0\rangle_{\mathbf{l}}$ (triplet state $|1\rangle_{\mathbf{l}}$) of the lth dimer of the original Shastry-Sutherland model (1). In addition to the occupation number operator $n_{\mathbf{l}}$ we have also introduced the creation and annihilation hard-core boson operators $a^+_{\mathbf{l}}$ and $a_{\mathbf{l}}$, which describe the transformation of the lth dimer from the singlet state $|0\rangle_{\mathbf{l}}$ to the triplet state $|1\rangle_{\mathbf{l}}$ and vice versa. The physical meaning of individual terms entering into the effective Hamiltonian (6) are as follows: $H_1$ corresponds to the renormalized energy of a single triplon, $H_2$ describes effective pair interactions between triplons placed on further-neighbor dimers (see Fig. 4), $H_3$ contains the most valuable contributions among effective three-particle interactions between triplons (see Fig. 5), and $H_t$ represents the correlated hopping term (see Fig. 6). The correlated hopping parameter $t$, the single-triplon energy term $e_0$, and all effective interaction potentials $V$'s emergent in Eq. (6) are analytic functions of $J$, $J'$ and $h$

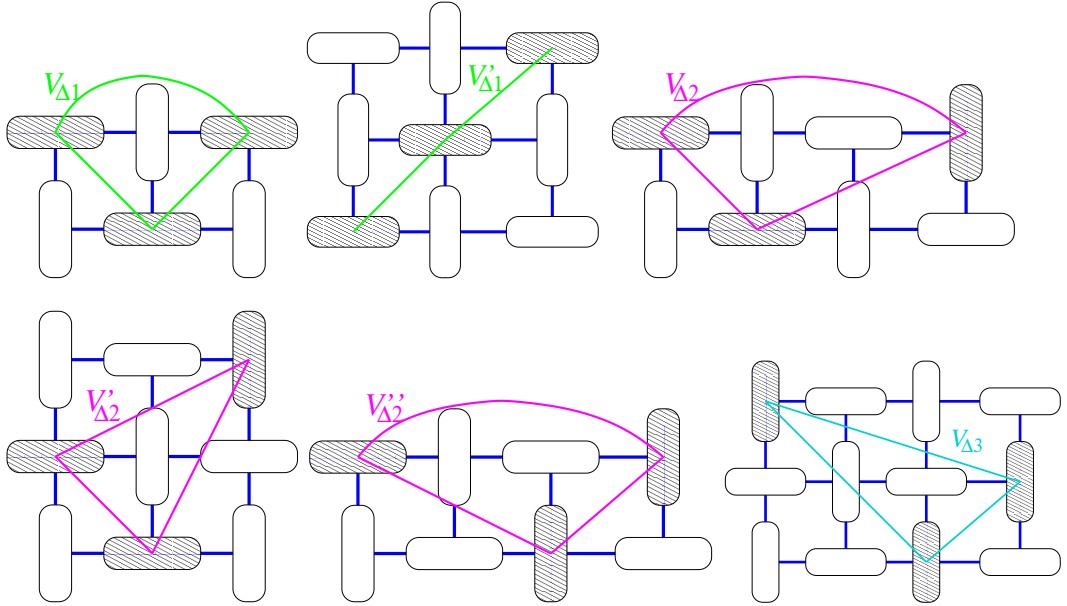

Figure 5: A schematic representation of effective three-particle interactions between triplons.

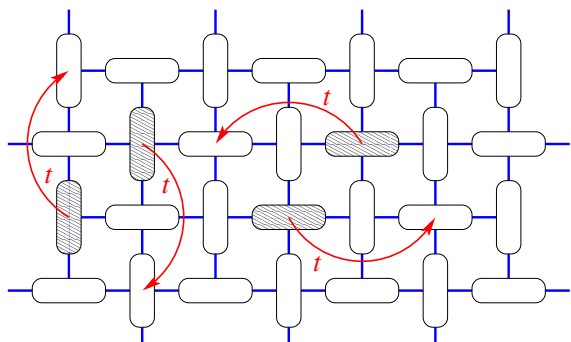

Figure 6: A schematic representation of the correlated hopping of two triplons, which are placed on two next-nearest-neighbor dimers. Arrows indicate the possible hoppings of triplons within a pair.

as dictated by the explicit expressions given in Appendix B.

Up to the second-order expansion, there are no direct tunneling terms for triplons. It should be mentioned, however, that such terms are negligibly small also in the ordinary strong-coupling approach starting from noninteracting dimers, where they appear only in the 6th order of the perturbation expansion [55]. In contrast to the effective Hamiltonian (6) the previous perturbative theories [13, 14, 56] include also the correlated hopping terms related to the configurations with triplons on the nearest horizontal and vertical dimers. As a consequence, the quantum state can be extended onto the whole lattice. We should mention that in the developed perturbation theory (6) the states with triplons on the nearest horizontal and vertical states are forbidden due to the hard-core condition schematically illustrated in Fig. 3. Therefore, the aforementioned correlated hopping terms will not appear even within the higher-order perturbation of our approach. For completeness, the coupling constants with a significant impact on a magnetic behavior entering in the effective Hamiltonian (6) are explicitly given in Appendix B, whereas their respective dependencies on the ratio of the coupling constants $J'/J$ are shown in Fig. 7. In general, the magnitudes of the effective couplings are relatively small for all values of the coupling ratio $J'/J$, which stabilize the singlet-dimer phase

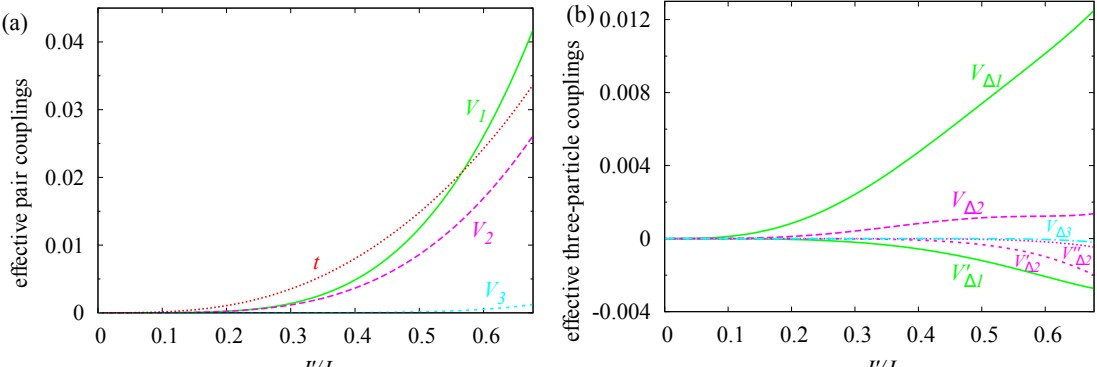

Figure 7: The effective pair (a) and three-particle (b) couplings (6) normalized with respect to the intradimer interaction $J$ as a function of the ratio of the coupling constants $J'/J$. The correlated hopping term is shown in panel (a) as a dotted line together with the effective pair couplings.

at zero magnetic field. Note furthermore that the three effective pair interactions generally decay with distance between dimers. Surprisingly, the correlated hopping parameter prevails at sufficiently small values of the interaction ratio $J'/J$, while the effective pair interaction $V_1$ between closest-spaced triplons allowed by the hard-core constraint (Fig. 3 and Eq. (B.4)) becomes the most dominant at higher values of the coupling ratio $J'/J$. It is evident from a comparison of Fig. 7(a) and (b) that the effective three-particle couplings are much smaller in magnitude than the effective pair interactions and thus, they do not have an essential impact on phase boundaries between different ground states. Moreover, the strength of the effective three-particle couplings drops off rapidly with the distance between triplons. It should be noted that the complete effective Hamiltonian developed up to the second order would also contain effective many-particle interactions of higher order (e.g. four-, five-, six-particle couplings). However, their values are expected to be much smaller in comparison with the effective three-particle interactions. This conjecture is based on the observation that the effective three-particle interactions are much (order of magnitude) smaller than the pair ones, and their values decay fast with the distance between triplons. Below, we show that the three-particle interaction results in small corrections to the transition field between different phases. Hence, the higher-order effective interactions are not envisaged to have any essential impact and we have therefore left out all those terms from the effective Hamiltonian (6).

As a first step we have found the ground-state phase diagram of the spin-1/2 Heisenberg model on the Shastry-Sutherland lattice by ignoring the correlated hopping terms, which might seem to be irrelevant due to the strong repulsion between closely spaced triplons. The analysis of the latter quantum terms is postponed till the next section. Thus, the problem becomes equivalent to finding the lowest-energy triplon configuration of the effective Hamiltonian (6) without the correlated hopping term. It can be proved that the ground state contains the phases corrsponding to the fractional magnetization plateaux 1/8, 1/6 and 1/4 (see Appendix C).

The first fractional magnetization plateau, which appears while applying the magnetic field, is the 1/8-plateau. The relevant ground state corresponds to the most dense packing of triplons, which avoids any repulsive pair and three-particle interactions between them. The 1/8-plateau phase corresponds to the highly degenerate ground-state manifold. The simplest columnar and checkerboard ordering of the vertical triplons are shown in Fig. 8 (a) and (b), respectively. As it is shown in Appendix C, the most general configuration can be obtained when the phases with vertical and horizontal ordering of triplons are mixed (see Fig. 17 and the discussions in Appendix C).

Upon further increase of the magnetic field the 1/6-plateau phase becomes favorable. The relevant ground state displays a striking columnar orderings of triplons developed either on

the horizontal or vertical dimers. The latter configuration is schematically shown in Fig. 8(c) for illustration. The degeneracy of the ground state at the border between the 1/8- and 1/6- plateau phases is an important feature. In Appendix C we show that the energy of the configurations is invariant under some local changes of the triplon configuration, e.g. one triplon can be replaced by two neighboring triplons as in Fig. 18(a). Therefore, any magnetization between the 1/8 and 1/6 magnetization can be achieved by such kind of substitution. In particular, the 2/15-plateau phase, which was identified in Refs. [6,7] and also revealed experimentally in Ref. [28], can be recovered here (see Fig. 18(b)) as one of the many coexisting states at the respective phase boundary. Hence, it is quite plausible to conjecture that the 2/15-plateau phase may eventually emerge when the perturbation expansion would be developed to higher orders.

The next consecutive ground state is the 1/4-plateau phase, which exhibits a stripe-like arrangements of triplons either on vertical or horizontal dimers. Fig. 8(d) illustrates the particular case, where triplons have vertical disposition. Note that the stripe-like character of the 1/4-plateau phase is ultimately connected to the effective three-particle interactions $V_{\triangle 1}$ and $V'_{\triangle 1}$, because the zigzag pattern of triplons has the same energy as the stripe one whenever the effective three-particle couplings $V_{\triangle 1}$ and $V'_{\triangle 1}$ are neglected. On the other hand, the effective three-particle interactions are much weaker than the pair ones and thus, the zigzag configuration mentioned above may emerge even at comparably small temperatures.

The last possible ground state within this picture is the 1/3-plateau phase schematically illustrated in Fig. 3, which displays another stripe ordering of triplons being simultaneously the most dense packing of triplons satisfying the hard-core condition exemplified by blue shaded region there. It could be thus concluded that the present consideration of the effective couplings between triplons leads to the emergence of three additional ground states emergent in between the singlet-dimer and stripe 1/3-plateau phases (see Fig. 8).

The ground-state energies of all aforementioned phases per one dimer read as follows:

$$E_{1/8} = \frac{1}{8}(-h + h_{c1} + e_0),$$

$$E_{1/6} = \frac{1}{6}(-h + h_{c1} + e_0 + 2V_3),$$

$$E_{1/4} = \frac{1}{4}(-h + h_{c1} + e_0 + V_1 + V_3 + V'_{\triangle 1} + 2V_{\triangle 3}),$$

$$E_{1/3} = \frac{1}{3}(-h + h_{c1} + e_0 + V_1 + 2V_2 + V'_{\triangle 1} + 2(V_{\triangle 2} + V'_{\triangle 2} + V''_{\triangle 2})). \tag{7}$$

It should be noted that all fractional-plateau phases are exact eigenstates of the effective Hamiltonian (6). The 1/8- and 1/6-plateau phases contain the localized triplons only. On the other hand, the correlated hopping of triplons placed on next-nearest-neighbor dimers (see Fig. 6) is hypothetically possible in the 1/4- and 1/3-plateau phases although it is still suppressed due to a stripe arrangement of triplons satisfying the hard-core condition sketched in Fig. 4. The critical fields delimiting phase boundaries between individual ground states can be readily found from a direct comparison of the respective eigenenergies given by Eqs. (7):

$$h_{0-1/8} = h_{c1} + e_0,$$

$$h_{1/8-1/6} = h_{c1} + e_0 + 8V_3,$$

$$h_{1/6-1/4} = h_{c1} + e_0 + 3V_1 - V_3 + 3V'_{\triangle 1} + 6V_{\triangle 3},$$

$$h_{1/4-1/3} = h_{c1} + e_0 + V_1 + 8V_2 - 3V_3 + V'_{\triangle 1} + 8(V_{\triangle 2} + V'_{\triangle 2} + V''_{\triangle 2}) - 6V_{\triangle 3}. \tag{8}$$

The critical fields (8) derived within the developed perturbation theory can be straightforwardly utilized for a construction of the overall ground-state phase diagram of the spin-1/2 Heisenberg model on the Shastry-Sutherland lattice, which is displayed in Fig. 9 in the

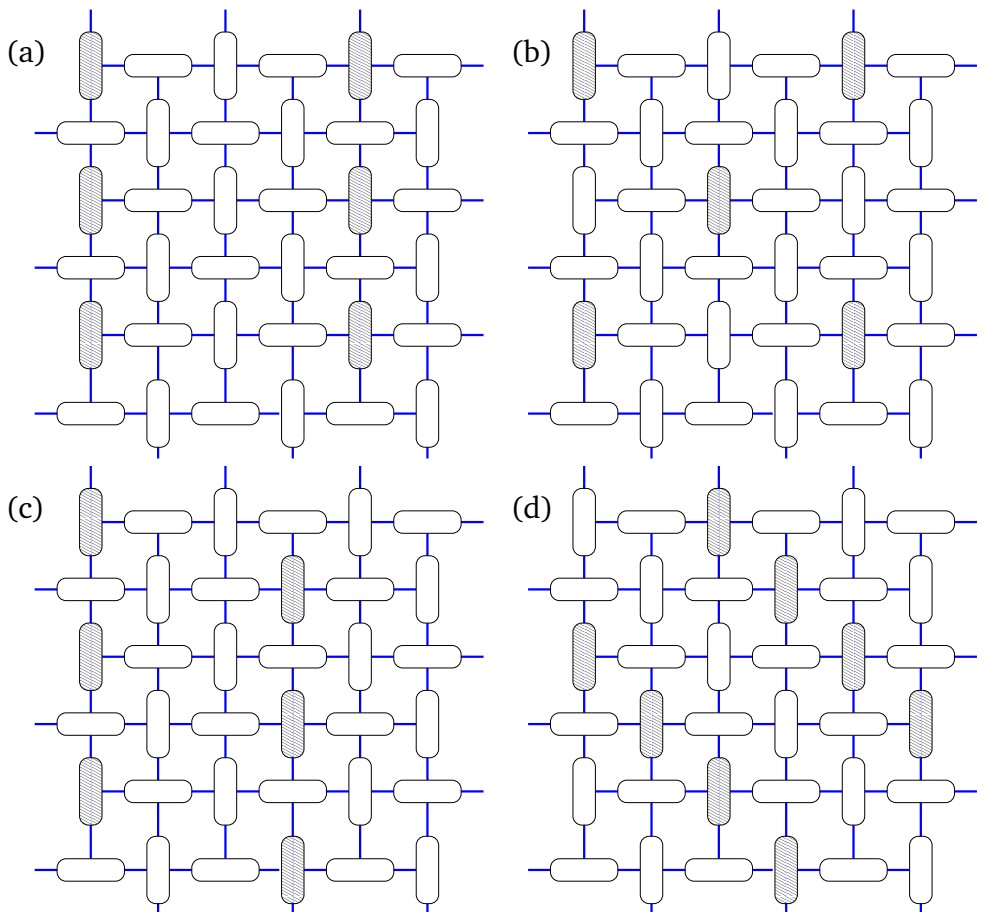

Figure 8: A schematic illustration of ground states of the spin-1/2 Heisenberg model on the Shastry-Sutherland lattice emergent in a low-field region as fractional magnetization plateaux: (a)-(b) stripe and checkerboard configuration for 1/8-plateau [57] ( see Appendix C for other degenerate configurations); (c) 1/6-plateau; (d) 1/4-plateau.

$J'/J - h/J$ plane together with available numerical data obtained previously using CORE [5] and iPEPS [10] methods. A direct comparison of the critical field $h_{1/4-1/3}$ with the respective numerical data of CORE [5] and iPEPS [10] methods implies that the phase boundary between the 1/4- and 1/3-plateaux is reproduced by the newly developed perturbation scheme with an exceptional high accuracy up to the interaction ratio $J'/J \approx 0.5$. The phase boundary between 1/6- and 1/4-plateaux is also in a reasonable accordance with the result of the iPEPS method [10] even up to higher values of the interaction ratio $J'/J \approx 0.6$. The deviation of our results from the CORE method [5] for the phase boundary between the 1/6- and 1/8-plateaux might be explained by the finite-size limitations. The developed strong-coupling approach anticipates a relatively broad field range for the 1/6-plateau phase and a tiny field range for the 1/8-plateau phase, whereas the 2/15-plateau coexists at the phase boundary between the 1/8- and 1/6-plateaux. However, this finding seems to be in contradiction with the experimental observation for $SrCu_2(BO_3)_2$ where a relatively broad 1/8-plateau was contrarily detected [27].

The obtained results open possibility to  understand to what extent the present results are able to describe the physical properties of $SrCu_2(BO_3)_2$ at low temperatures. It is clear from Fig. 9 that the phase boundaries of 1/4-plateau depend linearly on $J'/J$ near plausible values of the interaction ratio $J'/J \approx 0.6$. Using the linear approximation for these phase boundaries and experimentally observed fields $\mu_0 H_1^{exp} = 33$ T and $\mu_0 H_2^{exp} = 39$ T, which de-

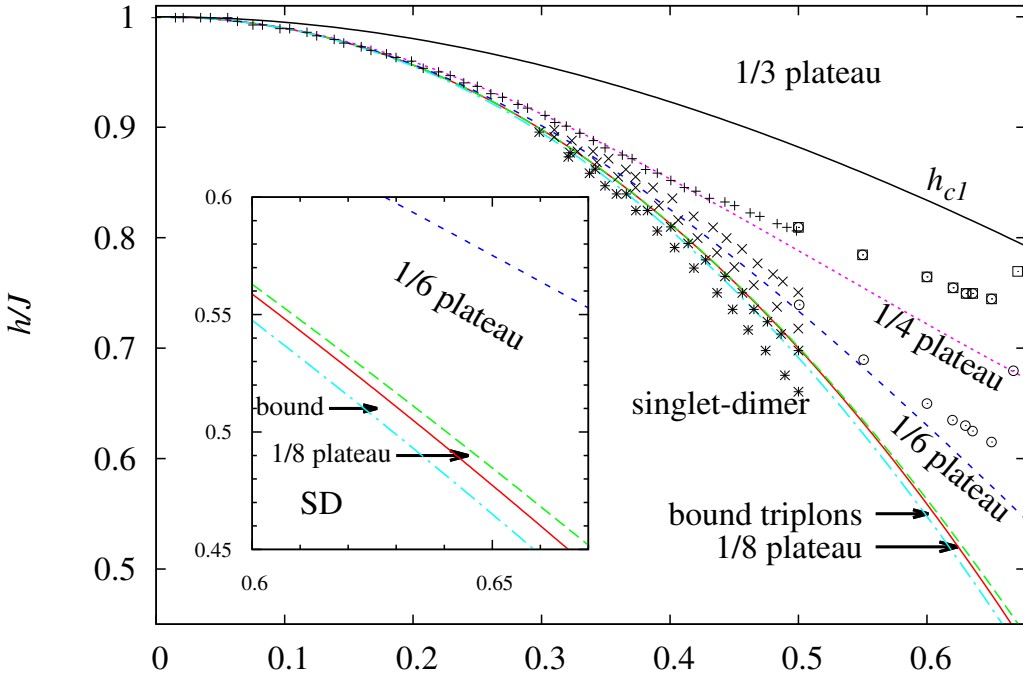

Figure 9: The ground-state phase diagram of the spin-1/2 Heisenberg model on the Shastry-Sutherland lattice in the $J'/J - h/J$ plane. The critical fields (8) derived within the perturbation theory, which was developed from the exactly solved Ising-Heisenberg model up to the second order, are shown by lines of different styles: $h_{0-1/8}$ (red solid curve), $h_{1/8-1/6}$ (green dash curve), $h_{1/6-1/4}$ (blue short-dash curve), $h_{1/4-1/3}$ (magenta dotted curve). The lowest dash-dotted line corresponds to the critical field $h_{bound-t}$ (12) discussed in Sec. 4. Symbols "+" display the phase boundary between the 1/4- and 1/3-plateaux, while the field range delimited by symbols "×" corresponds to the 1/6-plateau (for $N = 36$ spins) and the field range delimited by symbols "∗" corresponds to the 1/8-plateau (for $N = 32$ spins) obtained by the numerical CORE method [5]. Empty squares and circles show iPEPS results for lower critical fields of the 1/3- and 1/4-plateaux, respectively [10]. The inset shows an enlarged scale of the phase diagram for the $J'/J$ interval close to the presumable microscopic parameters for $SrCu_2(BO_3)_2$.

termine lower bounds of the 1/4- and 1/3-plateaux of $SrCu_2(BO_3)_2$, one finds absolute values of the coupling constants $J/k_B = 85.4$ K and $J'/k_B = 54.1$ K. Note that these specific values of the coupling constants as well as a relative strength of the interaction ratio $J'/J = 0.634$ are in a close coincidence with the previously reported fitting set of parameters gained from the temperature dependence of the magnetic susceptibility [58]. Next, the deduced coupling constants envisage for the phases with character of localized triplons the following critical field $\mu_0 H_{1/6-1/4} = 33$ T between the 1/6- and 1/4-plateaux and $\mu_0 H_{1/8-1/6} = 28$ T between the 1/8- and 1/6-plateaux, respectively. The energy of the localized triplon in zero field is found to be 43 K, which is slightly higher than the energy gap $\Delta/k_B = 35$ K experimentally observed at zero magnetic field [57]. The complete quantum spin model for $SrCu_2(BO_3)_2$ is however much more complex and it includes among other matters the weak Dzyaloshinskii-Moriya term [21,37,38] and possibly small interplane [59] couplings.

# 4 Quantum correlated phase of bound triplons

The presence of the quantum hopping term and the extended hard-core repulsion make the solution of the full effective model (6) including the correlated hopping rather complicated. The mean-field approach is not capable of providing a proper description of the quantum motion of triplons with strong short-order bonding. In our case the correlated hopping could be blocked by the strong repulsion between two triplons on next-nearest-neighbor dimers. On the other hand, a pair of bound triplons may gain even a lower energy due to the quantum correction terms.

The only rigorous result in this section is limited to the instability point where the energy of the single delocalized bound-triplon state becomes less than the energy of two singlet dimers. In this case, the bound triplons start to populate spin dimers leading to a complex quantum phase. At first, let us denote a state of bound horizontal triplons at the lattice positions $(i, j)$ and $(i+1, j+1)$ as $|1_{i,j} 1_{i+1,j+1}\rangle$. The action of the effective Hamiltonian (6) can be straightforwardly calculated as

$$H_{eff}|1_{i,j}1_{i+1,j+1}\rangle = [2(e_0+h_{c1}-h)+V_1]|1_{i,j}1_{i+1,j+1}\rangle + t(|1_{i-1,j+1}1_{i,j}\rangle + |1_{i+1,j+1}1_{i+2,j}\rangle). \quad (9)$$

It is clear that the motion of a triplon pair is one-dimensional, i.e. bound triplons on the horizontally (vertically) oriented dimers are moving in the horizontal (vertical) direction. Therefore, it is convenient to introduce the further notation for the bound-triplon state: $|\tilde{1}_i\rangle = |1_{i,j}1_{i+1,j+1}\rangle$, $|\tilde{1}_{i+1}\rangle = |1_{i+1,j+1}1_{i+2,j}\rangle$, and so on. Here we preserve only the index corresponding to the direction of the triplons movement. Now we can write the equation for the eigenenergies of the single bound-triplon excitation in a quite simple form

$$[2(e_0+h_{c1}-h)+V_1]|\tilde{1}_i\rangle + t(|\tilde{1}_{i-1}\rangle + |\tilde{1}_{i+1}\rangle) = E|\tilde{1}_i\rangle. \quad (10)$$

Implying the periodic boundary conditions, the solution of the difference equation has the form of the free-wave state $|\phi_k\rangle = \sum_l \exp(ikl)|\tilde{1}_l\rangle$ with the eigenspectrum

$$\epsilon(\kappa) = 2t\cos(\kappa) + 2(e_0 + h_{c1} - h) + V_1, \quad (11)$$

where $\kappa = 2\pi l/N_x$ and $l = 0, 1, \cdots, N_x$ ($N_x$ is the number of dimers in the horizontal direction). Of course, the energy spectrum and eigenstate of the vertically oriented triplons has an analogous form. Hence, it might be useful to compare the minimal energy of the bound-triplon state $\epsilon(\pi) = -2t + 2(e_0 + h_{c1} - h) + V_1$ with the energy of two separate (noninteracting) triplons $2E_{triplon} = 2(e_0 + h_{c1} - h)$. The difference of two energies $\Delta\varepsilon = \epsilon(\pi) - 2E_{triplon}$ is displayed in Fig. 10, which shows that the free-wave state of the bound triplons always has lower energy than a pair of localized triplons. Owing to this fact, the quantum phase of bound triplons should appear at low magnetic fields prior to the crystal of localized triplon phases. The critical field for the emergence of the quantum bound-triplon phase at sufficiently low magnetic fields is given by

$$h_{bound-t} = e_0 + h_{c1} - t + V_1/2. \quad (12)$$

The relevant phase boundary of the quantum bound-triplon phase is plotted in the ground-state phase diagram together with all other phase boundaries (see also inset in Fig. 9). It is quite evident that the energy of the quantum state of bound triplons rises rapidly with increasing their numbers, and therefore the stability region of such a quantum phase is limited to a very narrow range of magnetic fields. Moreover, the behavior of this quantum phase at larger magnetic fields is highly uncertain and would require a more comprehensive study.

To get some notion about the origin of the quantum phase of bound triplons, we consider a simplified model, where all states of localized triplons are excluded leaving only the possibility

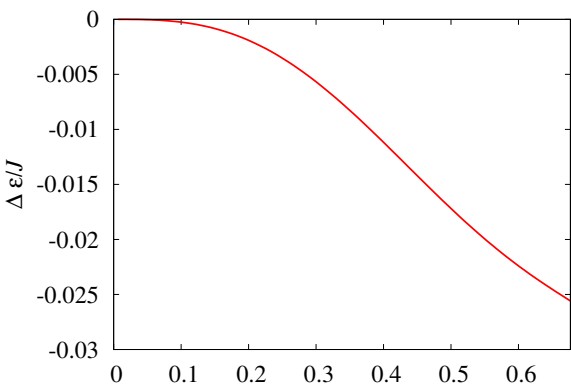

Figure 10: The energy difference between the bound triplons and two localized triplons as a function of their interaction ratio $J'/J$.

for the mobile pairs of triplons. For such a model one can devise the correspondence to the two-component hard-core Bose gas on the dual lattice (see Fig. 11) whose sites are defined at the center of pairs of dimers. Each site can be either empty or occupied by one of two kinds of hard-core bosons: $h$-bosons (or $v$-bosons) representing a bound pair of triplons moving along the horizontal (or vertical) direction. These bosons obey the following hard-core conditions: first, no more than one boson is allowed on a site, second, each boson on site $(i, j)$ forbids the occupation of neighboring sites according to the hard-core condition of separate triplons given by Eq. (B.4) and shown in Fig. 3 (see also shaded area in Fig. 11). These conditions can be fulfilled by the following projection operators for $h$-bosons:

$$
\begin{aligned}
\mathcal{P}_{i,j,h} &= (1 - n_{i,j,h} n_{i,j,v}) \overline{n}_{i-1,j-1} \overline{n}_{i,j-1} \overline{n}_{i+1,j-1} \overline{n}_{i+1,j} \overline{n}_{i+1,j+1} \overline{n}_{i,j+1} \overline{n}_{i-1,j+1} \overline{n}_{i-1,j} \\
&\times \overline{n}_{i-1,j-2} \overline{n}_{i,j-2} \overline{n}_{i,j+2} \overline{n}_{i+1,j+2} \overline{n}_{i-1,j-3,h} \overline{n}_{i,j-3,h} \overline{n}_{i+1,j-2,h} \\
&\times \overline{n}_{i-1,j+2,h} \overline{n}_{i,j+3,h} \overline{n}_{i+1,j+3,h} \overline{n}_{i-2,j-1,v} \overline{n}_{i-2,j,v} \overline{n}_{i+2,j,v} \overline{n}_{i+2,j+1,v}, \text{ if } i+j = \text{odd}, \\
\mathcal{P}_{i,j,h} &= (1 - n_{i,j,h} n_{i,j,v}) \overline{n}_{i-1,j-1} \overline{n}_{i,j-1} \overline{n}_{i+1,j-1} \overline{n}_{i+1,j} \overline{n}_{i+1,j+1} \overline{n}_{i,j+1} \overline{n}_{i-1,j+1} \overline{n}_{i-1,j} \\
&\times \overline{n}_{i,j-2} \overline{n}_{i+1,j-2} \overline{n}_{i-1,j+2} \overline{n}_{i,j+2} \overline{n}_{i-1,j-2,h} \overline{n}_{i,j-3,h} \overline{n}_{i+1,j-3,h} \\
&\times \overline{n}_{i-1,j+3,h} \overline{n}_{i,j+3,h} \overline{n}_{i+1,j+2,h} \overline{n}_{i-2,j,v} \overline{n}_{i-2,j+1,v} \overline{n}_{i+2,j-1,v} \overline{n}_{i+2,j,v}, \text{ if } i+j = \text{even}, \\
\overline{n}_{i,j,v} &= 1 - n_{i,j,v}, \quad \overline{n}_{i,j} = \overline{n}_{i,j,h} + \overline{n}_{i,j,v}.
\end{aligned} \tag{13}
$$

The projection operators for $v$-bosons $\mathcal{P}_{i,j,v}$ can be deduced from Eq. (13) by interchanging indices $i$ and $j$. Since triplons in a pair can move along its orientation only, the corresponding quasiparticle is able to move in the same direction. Now, one can see that the hopping of a triplon between sites can be presented as a jump of a quasiparticle identified on the dual lattice. Finally, one arrives at the following Hamiltonian:

$$
\begin{aligned}
H_{bound} = \mathcal{P}_{bt} \sum_{i,j} \Big\{ & t(b_{i,j,h}^+ b_{i+1,j,h} + b_{i+1,j,h}^+ b_{i,j,h} + b_{i,j,v}^+ b_{i,j+1,v} + b_{i,j+1,v}^+ b_{i,j,v}) \\
& + (2(e_0 + h_1 - h) + V_1) \sum_{\alpha=h,v} n_{i,j,\alpha} + (V_1 + 2V_{\triangle 1})(n_{i,j,h} n_{i+2,j,h} + n_{i,j,v} n_{i,j+2,v}) \\
& + (V_1 + V_{\triangle 1} + V_{\triangle 1}')(n_{i,j,h} n_{i+2,j+1,h} + n_{i,j,h} n_{i+2,j-1,h} + n_{i,j,v} n_{i+1,j+2,v} \\
& + n_{i,j,v} n_{i-1,j+2,v}) \Big\} \mathcal{P}_{bt},
\end{aligned} \tag{14}
$$

where $b_{i_j,\alpha}^+$ ($b_{i_j,\alpha}$) creates (annihilates) a pair of triplons in a horizontal (vertical) direction ($\alpha = h, v$), $\mathcal{P}_{bt}$ denotes the projection where the hard-core conditions (13) for bound triplons are incorporated. For better illustration we have depicted in Fig. 11 a particular example of a many-particle state of bound triplons. The pair of bound triplons represented by $h$-boson

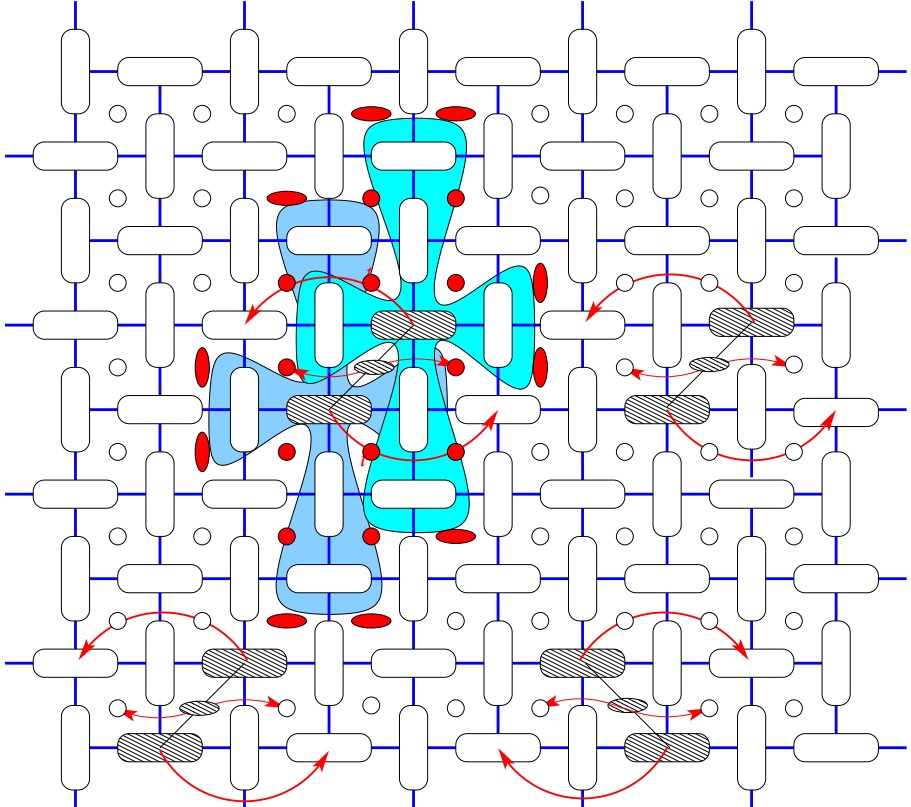

Figure 11: The schematic of the stripe-like phase for the effective model of bound triplons. Here, the same notations for dimers as in Fig. 3 are used. Small circles and ellipses denote sites of the dual lattice. The empty circles correspond to the empty sites, horizontally (vertically) oriented ellipses mark the bound state of two triplons in the horizontal (vertical) direction. The hard-core condition for bosons on the dual lattice is outlined by red circles (sites are forbidden for any boson) and red horizontal or vertical ellipse (site is forbidden for $h$- or $v$-boson). Red arrows indicate the possible hopping of triplons within a bound-triplon state, and the corresponding hopping of a triplon pair on a dual lattice.

creates a rather broad area of the hard-core repulsion for other bosons in the perpendicular direction with respect to its possible motion (see the shaded area in Fig. 11). On the other hand, the bosons along the direction of their hopping experience just the nearest-neighbor hard-core repulsion on the dual lattice.

It is still hard to get the exact solution even for the restricted model (14). However, we can look into the ground-state properties at magnetic fields close to $h_{bound-t}$ connected to appearance of bound triplons. At $h = h_{bound-t}$, the ground state is degenerate, since the energy of the single two-triplon bound state $\epsilon(\pi)$ (11) equals to zero. Bearing in mind the hard-core condition given by Eq. (13) and schematically shown in Fig. 11, we can deduce that the maximal number of the free-wave solutions for bound triplons is $N_y/4$, where each stripe (i.e. two rows) of the delocalized bound-triplon state should be separated by two rows of the singlet-like dimers. In case $h > h_{bound-t}$ it is not possible to accommodate a larger number of free-wave states of the bound triplons anymore. It is plausible to conjecture that the new bound triplon states will be created on the stripes already occupied by the free-wave states of bound triplons as it is shown in Fig. 11. Such an option allows to minimize the effect of the hard-core condition in contrast to the cases when an additional excitation is accommodated on the singlet-like stripes in-between two bound-triplon states, or across them which may

cost even much more energy. Such a picture is reminiscent of the 1/8-plateau phase revealed in the Shastry-Sutherland model on a tube geometry with the four dimers in the transverse direction [8]. In the latter case it is implemented as a wheel state of bound triplons. It is natural to suppose that by extending the width of the tube to infinity this state become identical to the stripe-phase of delocalized bound triplons suggested above.

## 5   Conclusions

In the present paper we have elaborated an approximate method for the spin-1/2 Heisenberg model on the Shastry-Sutherland lattice, which relies on the perturbative treatment of $XY$ part of the interdimer coupling. The spin-1/2 Ising-Heisenberg model on the Shastry-Sutherland lattice has been used as a useful starting ground (unperturbed reference system), for which the ground state is known exactly [52]. Our particular attention has been focused on a range of sufficiently low magnetic fields, which magnetize the system up to the intermediate 1/3-plateau when the magnetization is scaled with respect to the saturation value. Within the second order of the many-body perturbation theory, we have obtained an effective lattice-gas model for the triplon excitations with the special hard-core repulsion. This effective model allows for the consistent analytical description of the sequence of fractional 1/8-, 1/6- and 1/4-plateaux observed also in the related magnetic compound $SrCu_2(BO_3)_2$ [10,28]. The nature of ground states pertinent to these fractional magnetization plateaux was clarified in detail and it either corresponds to columnar or stripe orderings of localized triplons. Moreover, the stripe ordering of triplons in the 1/4-plateau phase is stabilized by a small three-particle interaction. Therefore, we showed explicitly that the $XY$ part of the interdimer coupling is responsible for the existence of smaller 1/8-, 1/6- and 1/4-plateaux. In addition, we have found that the ground state at the transition fields between the singlet-dimer and the 1/8-plateau phases, as well as between the 1/8- and 1/6-plateau phases are macroscopically degenerate. Higher-order perturbation theory can lift the macroscopic degeneracy, and hence one cannot rule out existence of other tiny plateaux in-between $0 - 1/8$ as well as $1/8 - 1/6$ plateaux. We have also analyzed the importance of the correlated hopping terms, the only quantum part of the effective Hamiltonian (6) which allows one to find the critical field as related to uprise of the quantum phase of bound triplon. Overall, we obtained a minimal effective model, which gives us the consistent picture for the magnetization plateaux in the Shastry-Sutherland model and clarifies their origin. The distinctive feature of the effective model is the one-dimensional character of the quantum hopping terms, which might imply the absence of the quantum frustration therein. This intriguing feature opens the possibility of more effective numerical simulations within the framework of the QMC method, which might bring insight into the manifestation of the quantum phase of bound triplons at finite temperatures even for rather large systems.

It is noteworthy that the derived critical fields for the fractional magnetization plateaux are in a good agreement with the available numerical data when the respective deviation is less than 5%. The linear dependence of the critical field near the presumed ratio of the coupling constants $J'/J \sim 0.6$ has allowed us to refine the coupling constants from the phase boundary of the 1/4-plateau observed experimentally in Refs. [10, 28]. A reliable agreement with the experimental data was also found when comparing the critical fields for other fractional magnetization plateaux with the ones detected in $SrCu_2(BO_3)_2$ [10, 27, 28]. In addition, it turns out that the calculated energy of the localized triplon excitations only slightly overestimates the energy gap of $SrCu_2(BO_3)_2$ experimentally observed in zero magnetic field [57]. Finally, the importance of quantum terms in the form of the correlated hopping process has been analyzed. It has been demonstrated that the correlated hopping may give rise to the creation of

the bound pair of triplons with much lower energy in comparison with two localized triplons. Thus, a stripe-like order of the delocalized bound triplons is presumed to evolve at sufficiently low magnetic fields. This conjecture might be of considerable interest for future experimental testing and numerical simulations.

## Acknowledgements

The authors are grateful to F. Mila and K.P. Schmidt for useful discussions.

**Funding information**  T.V. acknowledges the financial support provided by the National Scholarship Programme of the Slovak Republic for the Support of Mobility of Students, PhD Students, University Teachers, Researchers and Artists. J.S. acknowledges financial support provided by the grant from The Ministry of Education, Science, Research and Sport of the Slovak Republic under the contract No. VEGA 1/0105/20 and by the grant of the Slovak Research and Development Agency under the contract No. APVV-16-0186.

## A  Diagonalization of the Ising-Heisenberg model

In this part, we will derive the exact ground state of the spin-1/2 Ising-Heisenberg model on the Shastry-Sutherland lattice using the formalism of the projection operators [60, 61] $A_{i,j}^{ab} = |a\rangle_{i,j}\langle b|_{i,j}$, where $|a\rangle_{i,j}$ and $\langle b|_{i,j}$ are the dimer states defined by Eq. (3). For the sake of clarity, it is better to consider the relevant Hamiltonian (2) in the notation of full indices [52]:

$$H_{IH} = \sum_{i,j=1}^{N}{}' H_{i,[j-1:j+1]} + \sum_{i,j=1}^{N}{}'' H_{[i-1:i+1],j},$$

$$H_{[i-1:i+1],j} = J(\mathbf{s}_{1,i,j} \cdot \mathbf{s}_{2,i,j}) + J'(S_{i-1,j}^{z}s_{1,i,j}^{z} + s_{2,i,j}^{z}S_{i+1,j}^{z}) - hS_{i,j}^{z},$$

$$H_{i,[j-1:j+1]} = J(\mathbf{s}_{1,i,j} \cdot \mathbf{s}_{2,i,j}) + J'(S_{i,j-1}^{z}s_{1,i,j}^{z} + s_{2,i,j}^{z}S_{i,j+1}^{z}) - hS_{i,j}^{z}, \tag{A.1}$$

where $\sum'$ ($\sum''$) is restricted by the constraint $i+j=odd$ ($i+j=even$) and $S_{i,j}^{z} = s_{1,i,j}^{z} + s_{2,i,j}^{z}$ is the $z$-component of the total spin on a dimer. The cluster Hamiltonians (A.1) can be rewritten in terms of the introduced projection operators into the following form (see Ref. [62] for the similar representation of the spin-1/2 Ising-Heisenberg orthogonal-dimer chain):

$$H_{[i-1:i+1],j} = J\left(\frac{1}{4} - A_{ij}^{00}\right) - hS_{ij}^{z} + \frac{J'}{2}\left[(S_{i-1,j}^{z} + S_{i+1,j}^{z})S_{ij}^{z} + (S_{i-1,j}^{z} - S_{i+1,j}^{z})(A_{ij}^{20} + A_{ij}^{02})\right],$$

$$H_{i,[j-1:j+1]} = J\left(\frac{1}{4} - A_{ij}^{00}\right) - hS_{ij}^{z} + \frac{J'}{2}\left[(S_{i,j-1}^{z} + S_{i,j+1}^{z})S_{ij}^{z} + (S_{i,j-1}^{z} - S_{i,j+1}^{z})(A_{ij}^{20} + A_{ij}^{02})\right]. \tag{A.2}$$

For compactness, we have preserved the spin operator $S_{i,j}^{z} = A_{i,j}^{11} - A_{i,j}^{33}$ in the cluster Hamiltonians (A.2). It can be readily proved that all local Hamiltonians (A.2) commute with each other and, therefore, they can be diagonalized independently. Following Refs. [52,62] it is advisable to define the unitary transformation $U_{i,j}^{\nu}$ ($\nu = h, \nu$ denote either horizontal or vertical

orientation of the central dimer in a cluster):

$$U_{i,j}^{\nu} = (A_{i,j}^{11} + A_{i,j}^{33}) + \cos\frac{\alpha_{i,j}^{\nu}}{2}(A_{i,j}^{00} + A_{i,j}^{22}) + \sin\frac{\alpha_{i,j}^{\nu}}{2}(A_{i,j}^{20} - A_{i,j}^{02}),$$

$$\cos\alpha_{i,j}^{h} = \frac{J}{\sqrt{J^2 + J'^2(S_{i-1,j}^z - S_{i+1,j}^z)^2}}, \quad \sin\alpha_{i,j}^{h} = \frac{J'(S_{i-1,j}^z - S_{i+1,j}^z)}{\sqrt{J^2 + J'^2(S_{i-1,j}^z - S_{i+1,j}^z)^2}}, \text{ if } i+j=even,$$

$$\cos\alpha_{i,j}^{v} = \frac{J}{\sqrt{J^2 + J'^2(S_{i,j-1}^z - S_{i,j+1}^z)^2}}, \quad \sin\alpha_{i,j}^{v} = \frac{J'(S_{i,j-1}^z - S_{i,j+1}^z)}{\sqrt{J^2 + J'^2(S_{i,j-1}^z - S_{i,j+1}^z)^2}}, \text{ if } i+j=odd,$$

$$\cos\frac{\alpha_{i,j}^{h}}{2} = \sqrt{\frac{1+\cos\alpha_{i,j}^{h}}{2}} = 1 + (c_1^+ - 1)P_{|1|}(S_{i-1,j}^z - S_{i+1,j}^z) + (c_2^+ - 1)P_{|2|}(S_{i-1,j}^z - S_{i+1,j}^z),$$

$$\sin\frac{\alpha_{i,j}^{h}}{2} = \text{sgn}(\sin\alpha_{i,j}^{h})\sqrt{\frac{1-\cos\alpha_{i,j}^{h}}{2}} = c_1^- P_{\pm 1}(S_{i-1,j}^z - S_{i+1,j}^z) + c_2^- P_{\pm 2}(S_{i-1,j}^z - S_{i+1,j}^z),$$

$$\cos\frac{\alpha_{i,j}^{v}}{2} = 1 + (c_1^+ - 1)P_{|1|}(S_{i,j-1}^z - S_{i,j+1}^z) + (c_2^+ - 1)P_{|2|}(S_{i,j-1}^z - S_{i,j+1}^z),$$

$$\sin\frac{\alpha_{i,j}^{v}}{2} = c_1^- P_{\pm 1}(S_{i,j-1}^z - S_{i,j+1}^z) + c_2^- P_{\pm 2}(S_{i,j-1}^z - S_{i,j+1}^z),$$

$$c_1^{\pm} = \frac{1}{\sqrt{2}}\sqrt{1 \pm \frac{J}{\sqrt{J^2 + J'^2}}}, c_2^{\pm} = \frac{1}{\sqrt{2}}\sqrt{1 \pm \frac{J}{\sqrt{J^2 + 4J'^2}}},$$

$$P_{|1|}(S_{i,j}^z - S_{i',j'}^z) = \delta(|S_{i,j}^z - S_{i',j'}^z| - 1) = (A_{i,j}^{11} + A_{i,j}^{33})(A_{i',j'}^{00} + A_{i',j'}^{22}) + (A_{i,j}^{00} + A_{i,j}^{22})(A_{i',j'}^{11} + A_{i',j'}^{33}),$$

$$P_{|2|}(S_{i,j}^z - S_{i',j'}^z) = \delta(|S_{i,j}^z - S_{i',j'}^z| - 2) = A_{i,j}^{11}A_{i',j'}^{33} + A_{i,j}^{33}A_{i',j'}^{11},$$

$$\begin{aligned}P_{\pm 1}(S_{i,j}^z - S_{i',j'}^z) &= (S_{i,j}^z - S_{i',j'}^z)\delta(|S_{i,j}^z - S_{i',j'}^z| - 1)\\ &= (A_{i,j}^{11} - A_{i,j}^{33})(A_{i',j'}^{00} + A_{i',j'}^{22}) - (A_{i,j}^{00} + A_{i,j}^{22})(A_{i',j'}^{11} - A_{i',j'}^{33}),\end{aligned}$$

$$P_{\pm 2}(S_{i,j}^z - S_{i',j'}^z) = \frac{1}{2}(S_{i,j}^z - S_{i',j'}^z)\delta(|S_{i,j}^z - S_{i',j'}^z| - 2) = A_{i,j}^{11}A_{i',j'}^{33} - A_{i,j}^{33}A_{i',j'}^{11}, \tag{A.3}$$

where $\delta(\dots)$ denotes the Kronecker symbol. One can check that the unitary transformation $U_{i,j}^{\nu}$ removes all non-diagonal operators from the cluster Hamiltonians (A.2) what results in a "classical" (fully diagonal) representation of the Hamiltonian of the spin-1/2 Ising-Heisenberg model on the Shastry-Sutherland lattice [52]:

$$\begin{aligned}\tilde{H}_{[i-1:i+1],j} = U_{i,j}H_{[i-1:i+1],j}U_{i,j}^+ &= \left(\frac{J'}{2}(S_{i-1,j}^z + S_{i+1,j}^z) - h\right)S_{ij}^z + J\left(\frac{1}{4} - A_{ij}^{00}\right)\\ &+ \frac{1}{2}I(|S_{i-1,j}^z - S_{i+1,j}^z|)(A_{ij}^{22} - A_{ij}^{00}),\end{aligned}$$

$$\begin{aligned}\tilde{H}_{i,[j-1:j+1]} = U_{i,j}H_{i,[j-1:j+1]}U_{i,j}^+ &= \left(\frac{J'}{2}((S_{i,j-1}^z + S_{i,j+1}^z) - h\right)S_{ij}^z + J\left(\frac{1}{4} - A_{ij}^{00}\right)\\ &+ \frac{1}{2}I(|S_{i,j-1}^z - S_{i,j+1}^z|)(A_{ij}^{22} - A_{ij}^{00}),\end{aligned}$$

$$I(|S_{i+1,j}^z - S_{i-1,j}^z|) = \sqrt{J^2 + J'^2(S_{i+1,j}^z - S_{i-1,j}^z)^2} - J. \tag{A.4}$$

All ground states of the Hamiltonian (A.4) were found in Ref. [52], to which readers interested in further details are referred to. It should be stressed that 1/3 plateau can be found from the condition that only one out of three dimers can be excited into the triplet state within the three-dimer clusters given by Eq. (A.1) (see also Eq. (A.4) for the diagonal representation).

# B   Technical details of the perturbation theory

Let us start with the brief reference of the many-body perturbation theory adapted according to Ref. [54]. Within this scheme, it is convenient to decompose the total Hamiltonian into the unperturbed part $H_0$ and the perturbed part $H'$:

$$H = H_0 + \lambda H', \tag{B.1}$$

where the eigenvalue problem for the unperturbed part of the Hamiltonian $H_0|\Phi_i\rangle = E_i^{(0)}|\Phi_i\rangle$ should be rigorously tractable. If $\mathcal{P}_i$ is the projection into the unperturbed model space $H_0$ and $\mathcal{Q}_i = 1 - \mathcal{P}_i$, then, one gets within the perturbation theory:

$$H_{eff} = \mathcal{P}_i H \mathcal{P}_i + \lambda^2 \mathcal{P}_i H' R_s \sum_{n=0}^{\infty} \left[ (H' - \delta E_i) R_s \right]^n \mathcal{Q}_i H' \mathcal{P}_i,$$

$$R_s = \mathcal{Q}_i \frac{1}{E_i - H_0} = \sum_{m \neq i} \frac{|\Phi_m\rangle\langle\Phi_m|}{E_i - E_m^{(0)}} = \sum_{m \neq i} \frac{\mathcal{P}_m}{E_i - E_m^{(0)}}, \tag{B.2}$$

where $\delta E_i = E_i - E_i^{(0)}$ and $E_i$ is the energy eigenvalue corresponding the state $|\Phi_i\rangle$. Thus, one obtains from the perturbation expansion (B.2) the following effective Hamiltonians $H_{eff}^{(0)} = \mathcal{P}_0 H_0 \mathcal{P}_0$, $H_{eff}^{(1)} = \lambda \mathcal{P}_0 H' \mathcal{P}_0$, $H_{eff}^{(2)} = \lambda^2 \mathcal{P}_0 H' R_s H' \mathcal{P}_0$, where the effective Hamiltonian $H_{eff}^{(n)}$ denotes the $n$-th order of the perturbation expansion (B.2).

In particular, we will consider here the perturbation about the phase boundary between the singlet-dimer and stripe 1/3-plateau phase of the spin-1/2 Ising-Heisenberg model on the Shastry-Sutherland lattice emergent at the critical field $h_{c1} = -\sqrt{J^2 + J'^2} + 2J$ [52]. The ideal part $H^{(0)}$ of the Hamiltonian was chosen as the Ising-Heisenberg model (A.1) at the critical field $h_{c1}$, and the deviation from the critical field and the $XY$ part of the interdimer coupling goes into the perturbed part $H'$:

$$H' = \sum_{i,j=1}^{N}{}' H'_{i,j} + \sum_{i,j=1}^{N}{}'' H''_{i,j},$$

$$H'_{i,j} = \frac{J'_{xy}}{2} \left[ S_{i,j}^+ (s_{1,i,j-1}^- + s_{2,i,j-1}^-) + S_{i,j}^- (s_{1,i,j-1}^+ + s_{2,i,j-1}^+) \right] - (h - h_{c1}) S_{i,j}^z,$$

$$H''_{i,j} = \frac{J'_{xy}}{2} \left[ S_{i,j}^+ (s_{1,i+1,j}^- + s_{2,i-1,j}^-) + S_{i,j}^- (s_{1,i+1,j}^+ + s_{2,i-1,j}^+) \right] - (h - h_{c1}) S_{i,j}^z, \tag{B.3}$$

where we have introduced the spin raising and lowering operators $s_{l,i,j}^{\pm} = s_{l,i,j}^x \pm i s_{l,i,j}^y$, $H'_{i,j}$ and $H''_{i,j}$ correspond to the condition $i + j =$odd and $i + j =$even, respectively. Here, we have introduced the separate notation $J'_{xy}$ for the $XY$ part of the interdimer coupling in order to highlight the perturbation order, though this coupling constant will be finally put equal to the $z$ part of the interdimer coupling. As it was proved in Ref. [52], the ground state is the lattice gas of triplons with the hard-core repulsion given in Fig. 3. Henceforth, we will exploit the projection operators defined in Appendix A instead of the spin operators. The hard-core condition can be implemented by the following projection operator:

$$\mathcal{P}_0 = \prod_{i,j=1}^{N}{}' (A_{i,j}^{00} + A_{i,j}^{11} P_{ij}^h) \prod_{i,j=1}^{N}{}'' (A_{i,j}^{00} + A_{i,j}^{11} P_{ij}^v),$$

$$P_{ij}^h = A_{i,j-1}^{00} A_{i,j+1}^{00} A_{i-2,j}^{00} A_{i-1,j}^{00} A_{i+1,j}^{00} A_{i+2,j}^{00},$$

$$P_{ij}^v = A_{i-1,j}^{00} A_{i+1,j}^{00} A_{i,j-2}^{00} A_{i,j-1}^{00} A_{i,j+1}^{00} A_{i,j+2}^{00}. \tag{B.4}$$

It should be noted that the perturbation expansion needs to be performed for the unitary trans-formed operators $\tilde{O} = UOU^+$, where $U = \left(\prod' U_{i,j}\right)\left(\prod'' U_{i,j}\right)$. In this case, the Hamiltonian of the Ising-Heisenberg model gains a fully diagonal form as given by Eq. (A.4). The perturbation approach is valid for $J' < 0.678J$, where the zero-field ground state consists of a product of singlet dimers [3].

The first-order term is the perturbation operator projected on the subspace (B.4) $H^{(1)} = PUH'U^+P$. The field term gives the only contribution here:

$$H_{eff}^{(1)} = (h_{c1} - h)\sum_{i,j} A_{i,j}^{11}\mathcal{P}_0 \,. \tag{B.5}$$

The second-order term follows directly from Eq. (B.2):

$$H^{(2)} = \sum_{m\neq 0} \frac{\mathcal{P}_0 UH'U^+\mathcal{P}_m UH'U^+\mathcal{P}_0}{E_0^{(0)} - E_m^{(0)}} \,. \tag{B.6}$$

It is useful to be decompose the tedious calculation into more elemental parts. We start with the calculations of the following expressions explicitly $((i + j) = even)$:

$$Us_{1,i,j+1}^+ S_{i,j}^- U^+\mathcal{P}_0 = -(c_1^+ + c_1^-)A_{i,j}^{21}(c_1^+ A_{i,j-1}^{00} + c_1^- A_{i,j-1}^{20})\tilde{A}_{i,j+1}^{10}\mathcal{P}_0$$
$$+ c_1^-(A_{i-1,j}^{11} - A_{i+1,j}^{11})\tilde{A}_{i,j}^{30}\tilde{A}_{i,j+1}^{10}(A_{i,j+2}^{00} + (c_1^+ - c_1^-)A_{i,j+2}^{11})\mathcal{P}_0 \,,$$

$$Us_{2,i,j-1}^+ S_{i,j}^- U^+\mathcal{P}_0 = (c_1^+ + c_1^-)A_{i,j}^{21}(c_1^+ A_{i,j+1}^{00} + c_1^- A_{i,j+1}^{20})\tilde{A}_{i,j-1}^{10}\mathcal{P}_0$$
$$- c_1^-(A_{i-1,j}^{11} - A_{i+1,j}^{11})\tilde{\tilde{A}}_{i,j}^{30}\tilde{A}_{i,j-1}^{10}(A_{i,j-2}^{00} + (c_1^+ - c_1^-)A_{i,j-2}^{11})\mathcal{P}_0 \,,$$

$$\tilde{A}_{i,j}^{30} = A_{i,j}^{30}[A_{i,j-2}^{00}(A_{i,j-1}^{11} + c_1^+ A_{i,j-1}^{00} + c_1^- A_{i,j-1}^{20})$$
$$+ A_{i,j-2}^{11}\{(c_1^+ c_2^+ + c_1^- c_2^-)A_{i,j-1}^{00} + (c_1^+ c_2^- - c_1^- c_2^+)A_{i,j-1}^{20}\}] \,,$$

$$\tilde{\tilde{A}}_{i,j}^{30} = A_{i,j}^{30}[A_{i,j+1}^{11} + A_{i,j+2}^{00}(c_1^+ A_{i,j+1}^{00} + c_1^- A_{i,j+1}^{20})$$
$$+ A_{i,j+2}^{11}\{(c_1^+ c_2^+ + c_1^- c_2^-)A_{i,j+1}^{00} - (c_1^+ c_2^- - c_1^- c_2^+)A_{i,j+1}^{20}\}] \,,$$

$$\tilde{A}_{i,j\pm1}^{10} = A_{i,j\pm1}^{10}(A_{i-1,j\pm1}^{11} + c_1^+ A_{i-1,j\pm1}^{00} - c_1^- A_{i-1,j\pm1}^{20})(A_{i+1,j\pm1}^{11} + c_1^+ A_{i+1,j\pm1}^{00} - c_1^- A_{i+1,j\pm1}^{20}) \,, \tag{B.7}$$

$$Us_{1,i,j+1}^- S_{i,j}^+ U^+\mathcal{P}_0 = -c_1^- A_{i,j}^{10}(A_{i-1,j}^{11} - A_{i+1,j}^{11})(A_{i,j-1}^{11} + c_1^+ A_{i,j-1}^{00} - c_1^- A_{i,j-1}^{20})$$
$$\times \{(A_{i,j+2}^{00} + (c_1^+ + c_1^-)A_{i,j+2}^{11})\tilde{A}_{i,j+1}^{30} + [-(c_1^+ + c_1^-)A_{i,j+1}^{01} + (c_1^+ - c_1^-)A_{i,j+1}^{21}]$$
$$\times (c_1^+ A_{i-1,j+1}^{00} + c_1^- A_{i-1,j+1}^{20})(c_1^+ A_{i+1,j+1}^{00} - c_1^- A_{i+1,j+1}^{20})\}\mathcal{P}_0$$

$$Us_{2,i,j-1}^- S_{i,j}^+ U^+\mathcal{P}_0 = -c_1^- A_{i,j}^{10}(A_{i-1,j}^{11} - A_{i+1,j}^{11})(A_{i,j+1}^{11} + c_1^+ A_{i,j+1}^{00} + c_1^- A_{i,j+1}^{20})$$
$$\times \{-(A_{i,j-2}^{00} + (c_1^+ + c_1^-)A_{i,j-2}^{11})\tilde{A}_{i,j-1}^{30} + [(c_1^+ + c_1^-)A_{i,j-1}^{01} + (c_1^+ - c_1^-)A_{i,j-1}^{21}]$$
$$\times (c_1^+ A_{i-1,j-1}^{00} + c_1^- A_{i-1,j-1}^{20})(c_1^+ A_{i+1,j-1}^{00} - c_1^- A_{i+1,j-1}^{20})\}\mathcal{P}_0$$

$$\tilde{A}_{i,j\pm1}^{30} = A_{i,j\pm1}^{30}[A_{i+1,j\pm1}^{11} + A_{i+2,j\pm1}^{00}(c_1^+ A_{i+1,j\pm1}^{00} - c_1^- A_{i+1,j\pm1}^{20})$$
$$+ A_{i+2,j\pm1}^{11}\{(c_1^+ c_2^+ + c_1^- c_2^-)A_{i+1,j\pm1}^{00} - (c_1^+ c_2^- - c_1^- c_2^+)A_{i+1,j\pm1}^{20}\}]$$
$$\times [A_{i-1,j\pm1}^{11} + A_{i-2,j\pm1}^{00}(c_1^+ A_{i-1,j\pm1}^{00} + c_1^- A_{i-1,j\pm1}^{20})$$
$$+ A_{i-2,j\pm1}^{11}\{(c_1^+ c_2^+ + c_1^- c_2^-)A_{i-1,j\pm1}^{00} + (c_1^+ c_2^- - c_1^- c_2^+)A_{i-1,j\pm1}^{20}\}] \,, \tag{B.8}$$

where $c_1^\pm$ and $c_2^\pm$ are given in Eqs. (A.3). The expressions for $Us_{1,i+1,j}^+ S_{i,j}^- U^+\mathcal{P}_0$, $Us_{2,i-1,j}^+ S_{i,j}^- U^+\mathcal{P}_0$, $Us_{1,i+1,j}^- S_{i,j}^+ U^+\mathcal{P}_0$, $Us_{2,i-1,j}^- S_{i,j}^+ U^+\mathcal{P}_0$ can be recovered from the equations (B.7)-(B.8) by interchanging the indices.

The expressions (B.7)-(B.8) can be subsequently inserted into Eq. (B.6) in order to get the effective Hamiltonian in the second order. However, this is rather cumbersome combinatorial problem and we provide here only a sketch of such calculation. At first let us consider the diagonal terms, which appear by contracting the perturbation terms acting on the same sites:

$$
\begin{aligned}
H^{(2)} =& \frac{(J'_{xy})^2}{4} \sum_{i,j=1}^{N}{}' \sum_{m\neq 0} \Bigg( \frac{\mathcal{P}_0 U s^-_{1,i,j+1} S^+_{i,j} U^+ \mathcal{P}_m U s^+_{1,i,j+1} S^-_{i,j} U^+ \mathcal{P}_0}{E_0^{(0)} - E_m^{(0)}} \\
&+ \frac{\mathcal{P}_0 U s^+_{1,i,j+1} S^-_{i,j} U^+ \mathcal{P}_m U s^-_{1,i,j+1} S^+_{i,j} U^+ \mathcal{P}_0}{E_0^{(0)} - E_m^{(0)}} + \frac{\mathcal{P}_0 U s^-_{2,i,j-1} S^+_{i,j} U^+ \mathcal{P}_m U s^+_{2,i,j-1} S^-_{i,j} U^+ \mathcal{P}_0}{E_0^{(0)} - E_m^{(0)}} \\
&+ \frac{\mathcal{P}_0 U s^+_{2,i,j-1} S^-_{i,j} U^+ \mathcal{P}_m U s^-_{2,i,j-1} S^+_{i,j} U^+ \mathcal{P}_0}{E_0^{(0)} - E_m^{(0)}} \Bigg) \\
&+ \frac{(J'_{xy})^2}{4} \sum_{i,j=1}^{N}{}'' \sum_{m\neq 0} \Bigg( \frac{\mathcal{P}_0 U s^-_{1,i+1,j} S^+_{i,j} U^+ \mathcal{P}_m U s^+_{1,i+1,j} S^-_{i,j} U^+ \mathcal{P}_0}{E_0^{(0)} - E_m^{(0)}} \\
&+ \frac{\mathcal{P}_0 U s^+_{1,i+1,j} S^-_{i,j} U^+ \mathcal{P}_m U s^-_{1,i+1,j} S^+_{i,j} U^+ \mathcal{P}_0}{E_0^{(0)} - E_m^{(0)}} + \frac{\mathcal{P}_0 U s^-_{2,i-1,j} S^+_{i,j} U^+ \mathcal{P}_m U s^+_{2,i-1,j} S^-_{i,j} U^+ \mathcal{P}_0}{E_0^{(0)} - E_m^{(0)}} \\
&+ \frac{\mathcal{P}_0 U s^+_{2,i-1,j} S^-_{i,j} U^+ \mathcal{P}_m U s^-_{2,i-1,j} S^+_{i,j} U^+ \mathcal{P}_0}{E_0^{(0)} - E_m^{(0)}} \Bigg),
\end{aligned}
\tag{B.9}
$$

where $E_0^{(0)}$ is the energy of the singlet-dimer state, and $E_m^{(0)}$ are the energies of the states excited by the perturbation operator.

Further we will single out terms containing one, two or more triplet states separately. At first we extract the terms containing projection to the triplet excitation just on one site, while other sites are either in the singlet state or their states are not specified. It gives the one-particle contribution

$$
\begin{aligned}
H_1^{(2)} =& E_0 \sum_{i,j} A^{11}_{i,j} \mathcal{P}_0, \\
E_0 =& -\frac{(J'_{xy})^2}{4} \Bigg\{ 2\left[ \frac{(c_1^+ + c_1^-)^2 (c_1^+)^6}{J} + 2\frac{(c_1^+ + c_1^-)^2 (c_1^+)^4 (c_1^-)^2}{J + \sqrt{J^2 + J'^2}} + \frac{(c_1^+ + c_1^-)^2 (c_1^+)^4 (c_1^-)^2}{2J} \right] \\
&+ 4\left( \frac{(c_1^+)^6 (c_1^-)^2}{3J - J' - \sqrt{J^2 + J'^2}} + \frac{(c_1^+)^6 (c_1^-)^2}{3J - \sqrt{J^2 + J'^2}} \right) \Bigg\}.
\end{aligned}
\tag{B.10}
$$

Hereafter we retain the contributions up to $(c_1^-)^2$ ($(c_1^+)^2 \approx 0.91$, $(c_1^-)^2 \approx 0.09$ at the limiting value $J'/J = 0.7$). The contribution for the two-triplon interaction can be found by extracting from Eq. (B.9) the terms containing the product of projections on two triplon states. Additionally, we have to subtract the one-particle contributions due to the relation $A^{00}_{i,j} = 1 - A^{11}_{i,j}$. As a result we get:

$$
\begin{aligned}
H_2^{(2)} =& \Bigg\{ \sum_{i,j} \Bigg[ K_1 A^{11}_{i,j} \sum_{n_1=\pm 1} \sum_{n_2=\pm 1} A^{11}_{i+n_1,j+n_2} \\
&+ K_2 A^{11}_{i,j} \left( \sum_{n_1=\pm 2} \sum_{n_2=\pm 1} A^{11}_{i+n_1,j+n_2} + \sum_{n_1=\pm 1} \sum_{n_2=\pm 2} A^{11}_{i+n_1,j+n_2} \right) \Bigg] \\
&+ K_3 \left( \sum_{i,j=1}^{N}{}' A^{11}_{i,j} \sum_{n_1=\pm 3} \sum_{n_2=\pm 1} A^{11}_{i+n_1,j+n_2} + \sum_{i,j=1}^{N}{}'' A^{11}_{i,j} \sum_{n_1=\pm 1} \sum_{n_2=\pm 3} A^{11}_{i+n_1,j+n_2} \right) \Bigg\} \mathcal{P}_0,
\end{aligned}
\tag{B.11}
$$

where the effective parameters are as follows:

$$K_1 = -\frac{(J'_{xy})^2}{4}\left[\frac{2(c_1^+ + c_1^-)^2(c_1^+)^4}{J+J'+\sqrt{J^2+J'^2}} + \frac{2(c_1^+ + c_1^-)^2(c_1^+)^2(c_1^-)^2}{J+J'+3\sqrt{J^2+J'^2}} + \frac{2(c_1^+ + c_1^-)^2(c_1^+)^2(c_1^-)^2}{3J+J'+\sqrt{J^2+J'^2}}\right.$$

$$+\frac{2(c_1^+)^4(c_1^-)^2}{5J-3J'-\sqrt{J^2+J'^2}} + \frac{(c_1^+)^6(c_1^-)^2}{2J-J'} + \frac{2(c_1^+)^4(c_1^-)^2}{5J+J'-\sqrt{J^2+J'^2}}$$

$$+\frac{2(c_1^+ + c_1^-)^2(c_1^+)^6(c_1^-)^2}{-J+J'+\sqrt{J^2+J'^2}} + \frac{2(c_1^+ - c_1^-)^2(c_1^+)^6(c_1^-)^2}{-J+J'+3\sqrt{J^2+J'^2}} + \frac{2(c_1^+ c_2^+ + c_1^- c_2^-)^2(c_1^+)^2(c_1^-)^2}{5J-\sqrt{J^2+4J'^2}}$$

$$-\frac{(c_1^+ + c_1^-)^2(c_1^+)^6}{J} - 2\frac{(c_1^+ + c_1^-)^2(c_1^+)^4(c_1^-)^2}{J+\sqrt{J^2+J'^2}} - \frac{(c_1^+ + c_1^-)^2(c_1^+)^4(c_1^-)^2}{2J}$$

$$-3\left(\frac{(c_1^+)^6(c_1^-)^2}{3J-J'-\sqrt{J^2+J'^2}} + \frac{(c_1^+)^6(c_1^-)^2}{3J-\sqrt{J^2+J'^2}}\right)\Bigg],$$

$$K_2 = \frac{(J'_{xy})^2}{8}\left[\frac{(c_1^+ + c_1^-)^2(c_1^+)^6}{\sqrt{J^2+J'^2}} + \frac{(c_1^+ + c_1^-)^2(c_1^+)^4(c_1^-)^2}{J+\sqrt{J^2+J'^2}} + \frac{(c_1^+ + c_1^-)^2(c_1^+)^4(c_1^-)^2}{2\sqrt{J^2+J'^2}}\right.$$

$$+\frac{2(c_1^+)^4(c_1^-)^2}{5J-J'-\sqrt{J^2+J'^2}} + \frac{2(c_1^+ c_2^+ + c_1^- c_2^-)^2(c_1^+)^4(c_1^-)^2}{5J-2J'-\sqrt{J^2+4J'^2}} + \frac{2(c_1^+ - c_1^-)^2(c_1^+)^6(c_1^-)^2}{5J-J'-\sqrt{J^2+J'^2}}$$

$$+\frac{2(c_1^+)^4(c_1^-)^2}{5J-J'-\sqrt{J^2+J'^2}} + \frac{(c_1^+)^6(c_1^-)^2}{2J} + \frac{2(c_1^+ + c_1^-)^2(c_1^+)^6(c_1^-)^2}{5J-J'-\sqrt{J^2+J'^2}}$$

$$-\frac{(c_1^+ + c_1^-)^2(c_1^+)^6}{J} - 2\frac{(c_1^+ + c_1^-)^2(c_1^+)^4(c_1^-)^2}{J+\sqrt{J^2+J'^2}} - \frac{(c_1^+ + c_1^-)^2(c_1^+)^4(c_1^-)^2}{2J}$$

$$-3\left(\frac{(c_1^+)^6(c_1^-)^2}{3J-J'-\sqrt{J^2+J'^2}} + \frac{(c_1^+)^6(c_1^-)^2}{3J-\sqrt{J^2+J'^2}},\right)\Bigg],$$

$$K_3 = \frac{(J'_{xy})^2}{4}\left[\frac{(c_1^+)^6(c_1^-)^2}{2J-J'} + \frac{2(c_1^+ c_2^+ + c_1^- c_2^-)^2(c_1^+)^4(c_1^-)^2}{5J-\sqrt{J^2+4J'^2}} - \frac{(c_1^+)^6(c_1^-)^2}{3J-J'-\sqrt{J^2+J'^2}}\right.$$

$$\left. -\frac{(c_1^+)^6(c_1^-)^2}{3J-\sqrt{J^2+J'^2}}\right]. \tag{B.12}$$

The significant part of three-particle interactions includes the triplon configurations where at least two of them are at the closest location with respect to each other. It corresponds to the following effective Hamiltonian:

$$H_3^{(2)} = \left\{ K_{\triangle 1}\left(\sum_{i,j=1}^{N}{}' A_{i,j}^{11}\sum_{n=\pm 1} A_{i-1,j+n}^{11}A_{i+1,j+n}^{11} + \sum_{i,j=1}^{N}{}'' A_{i,j}^{11}\sum_{n=\pm 1} A_{i+n,j-1}^{11}A_{i+n,j+1}^{11}\right)\right.$$

$$+ K'_{\triangle 1}\sum_{i,j=1}^{N} A_{i,j}^{11}(A_{i-1,j-1}^{11}A_{i+1,j+1}^{11} + A_{i-1,j+1}^{11}A_{i+1,j-1}^{11})$$

$$+ K_{\triangle 2}\left[\sum_{i,j=1}^{N}{}' A_{i,j}^{11}\sum_{n=\pm 1}\left(A_{i-1,j+n}^{11}A_{i+2,j+n}^{11} + A_{i-2,j+n}^{11}A_{i+1,j+n}^{11}\right)\right.$$

$$\left. + \sum_{i,j=1}^{N}{}'' A_{i,j}^{11}\sum_{n=\pm 1}\left(A_{i+n,j-1}^{11}A_{i+n,j+2}^{11} + A_{i+n,j-2}^{11}A_{i+n,j+1}^{11}\right)\right]$$

$$+ K'_{\triangle 2}\sum_{i,j=1}^{N} A_{i,j}^{11}\sum_{n=\pm 1}\left(A_{i-2n,j-n}^{11}A_{i-n,j-2n}^{11} + A_{i+2n,j-n}^{11}A_{i+n,j-2n}^{11}\right)$$

$$+ K''_{\triangle 2} \left[ \sum_{i,j=1}^{N}{}' A_{i,j}^{11} \sum_{n=\pm 1} \left( A_{i-2n,j-n}^{11} A_{i-3n,j}^{11} + A_{i+2n,j-n}^{11} A_{i+3n,j}^{11} \right) \right.$$

$$+ \left. \sum_{i,j=1}^{N}{}'' A_{i,j}^{11} \sum_{n=\pm 1} \left( A_{i-n,j-2n}^{11} A_{i,j-3n}^{11} + A_{i-n,j+2n}^{11} A_{i,j+3n}^{11} \right) \right]$$

$$+ K_{\triangle 3} \left[ \sum_{i,j=1}^{N}{}' A_{i,j}^{11} \sum_{n=\pm 1} A_{i-2,j+n}^{11} A_{i+2,j+n}^{11} + \sum_{i,j=1}^{N}{}'' A_{i,j}^{11} \sum_{n=\pm 1} A_{i+n,j-2}^{11} A_{i+n,j+2}^{11} \right] \right\} \mathcal{P}_0. \quad \text{(B.13)}$$

We omit here the explicit expressions for the effective three-particle couplings because of their extensive length. However, the dependencies of these parameters are shown in Fig. 7(b).

The contribution of quantum terms is obtained by contracting the terms corresponding to neighboring interdimer couplings:

$$H^{(2)} = \frac{(J'_{xy})^2}{4} \sum_{i,j=1}^{N}{}' \sum_{m \neq 0} \left( \frac{\mathcal{P}_0 U s_{2,i,j-1}^{-} S_{i,j}^{+} U^{+} \mathcal{P}_m U s_{1,i,j+1}^{+} S_{i,j}^{-} U^{+} \mathcal{P}_0}{E_0^{(0)} - E_m^{(0)}} \right.$$

$$+ \frac{\mathcal{P}_0 U s_{2,i,j-1}^{+} S_{i,j}^{-} U^{+} \mathcal{P}_m U s_{1,i,j+1}^{-} S_{i,j}^{+} U^{+} \mathcal{P}_0}{E_0^{(0)} - E_m^{(0)}} + \frac{\mathcal{P}_0 U s_{1,i,j+1}^{-} S_{i,j}^{+} U^{+} \mathcal{P}_m U s_{2,i,j-1}^{+} S_{i,j}^{-} U^{+} \mathcal{P}_0}{E_0^{(0)} - E_m^{(0)}}$$

$$+ \left. \frac{\mathcal{P}_0 U s_{1,i,j+1}^{+} S_{i,j}^{-} U^{+} \mathcal{P}_m U s_{2,i,j-1}^{-} S_{i,j}^{+} U^{+} \mathcal{P}_0}{E_0^{(0)} - E_m^{(0)}} \right)$$

$$+ \frac{(J'_{xy})^2}{4} \sum_{i,j=1}^{N}{}'' \sum_{m \neq 0} \left( \frac{\mathcal{P}_0 U s_{2,i-1,j}^{-} S_{i,j}^{+} U^{+} \mathcal{P}_m U s_{1,i+1,j}^{+} S_{i,j}^{-} U^{+} \mathcal{P}_0}{E_0^{(0)} - E_m^{(0)}} \right.$$

$$+ \frac{\mathcal{P}_0 U s_{2,i-1,j}^{+} S_{i,j}^{-} U^{+} \mathcal{P}_m U s_{1,i+1,j}^{-} S_{i,j}^{+} U^{+} \mathcal{P}_0}{E_0^{(0)} - E_m^{(0)}} + \frac{\mathcal{P}_0 U s_{1,i+1,j}^{-} S_{i,j}^{+} U^{+} \mathcal{P}_m U s_{2,i-1,j}^{+} S_{i,j}^{-} U^{+} \mathcal{P}_0}{E_0^{(0)} - E_m^{(0)}}$$

$$+ \left. \frac{\mathcal{P}_0 U s_{1,i+1,j}^{+} S_{i,j}^{-} U^{+} \mathcal{P}_m U s_{2,i-1,j}^{-} S_{i,j}^{+} U^{+} \mathcal{P}_0}{E_0^{(0)} - E_m^{(0)}} \right). \quad \text{(B.14)}$$

For the sake of clarity, the diagrams on Fig. 12 illustrate the action of the perturbation terms in the second-order theory. The first two diagrams correspond to the contribution of the first term in Eq. (29), which can be reduced to the following form in terms of the projection operators:

$$\frac{(J'_{xy})^2}{4} \frac{(c_1^{-})^2 (c_1^{+})^4 \mathcal{P}(A_{i-1,j}^{11} - A_{i+1,j}^{11}) A_{i,j}^{03} A_{i,j+1}^{11} A_{i,j-1}^{01} \mathcal{P}_a^{*} A_{i,j}^{30} A_{i,j+1}^{10} A_{i,j-1}^{11} \mathcal{P}}{(E_0^{(0)} - E_a^{*})}. \quad \text{(B.15)}$$

The others diagrams represent the contributions proportional to $(c_1^{-})^2$ of the second term in Eq. (29) as the following result

$$\frac{(J'_{xy})^2}{4} \frac{(c_1^{-})^2 (c_1^{+})^6 (c_1^{+} + c_1^{-})^2 \mathcal{P}(A_{i-1,j}^{11} - A_{i+1,j}^{11}) A_{i,j}^{01} A_{i,j+1}^{00} A_{i,j-1}^{01} \mathcal{P}_b^{*} A_{i,j}^{10} A_{i,j+1}^{01} A_{i,j-1}^{00} \mathcal{P}}{(E_0^{(0)} - E_b^{*})}. \quad \text{(B.16)}$$

Here $\mathcal{P}_a^{*}, E_a^{*}$ and $\mathcal{P}_b^{*}, E_b^{*}$ are the projection and the energy of the excited states shown schematically on panels (a) and (b) of Fig. 12. Eq. (B.14) can be recast into the form involving the

(a) (b)

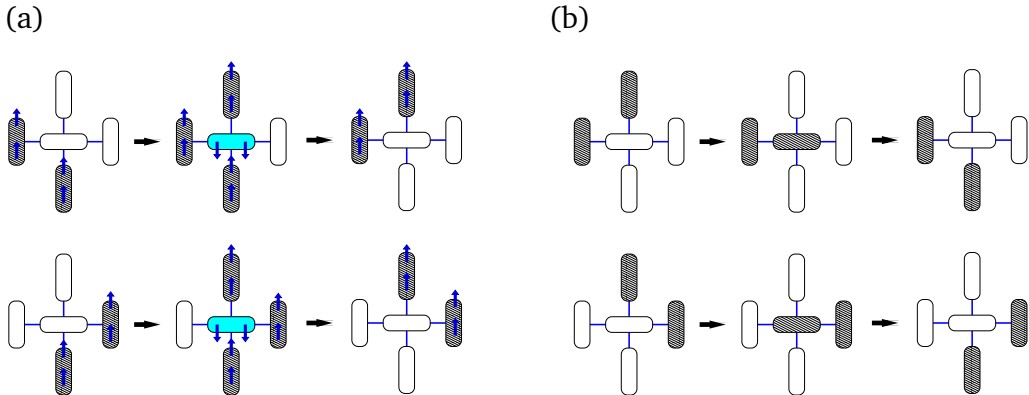

Figure 12: Diagrams presenting virtual hopping processes leading to the correlated hopping term in the effective Hamiltonian (6). Panels (a) and (b) correspond to Eqs. (B.15) and (B.16). The shaded ovals correspond to the triplons with $S^z_{i,j} = 1$ (black) or $S^z_{i,j} = -1$ (cyan).

correlated hopping of triplet excitations on a lattice:

$$
\begin{aligned}
H_t^{(2)} = &\sum_{i,j=1}^{N}{}' t(A^{11}_{i,j-1} + A^{11}_{i,j+1})(A^{10}_{i+1,j}A^{01}_{i-1,j} + A^{01}_{i+1,j}A^{10}_{i-1,j}) \\
&+ \sum_{i,j=1}^{N}{}'' t(A^{11}_{i-1,j} + A^{11}_{i+1,j})(A^{10}_{i,j+1}A^{01}_{i,j-1} + A^{01}_{i,j+1}A^{10}_{i,j-1}) \\
t = &\frac{(J'_{xy})^2}{4}(c_1^+)^4(c_1^-)^2 \left[ \frac{2}{5J - 3J' - \sqrt{J^2 + J'^2}} + \frac{2(c_1^+ + c_1^-)^2(c_1^+)^2}{-J + J' + \sqrt{J^2 + J'^2}} \right].
\end{aligned}
\tag{B.17}
$$

All second-order contributions given by Eqs. (B.10)-(B.14) are collected in the effective Hamiltonian:

$$
H_{eff}^{(2)} = H_0^{(2)} + H_1^{(2)} + H_t^{(2)} + H_2^{(2)} + H_3^{(2)} + \dots .
\tag{B.18}
$$

Instead of the projection operators we can introduce the representation of the quantum lattice gas, where $n_{i,j} = A^{11}_{i,j}$ is the occupation number operator, and $a^+_{i,j} = A^{10}_{i,j}$ ($a_{i,j} = A^{01}_{i,j}$) is the triplon creation (annihilation) operator. Thus, the effective Hamiltonian can be presented in a more compact form:

$$
\begin{aligned}
H_1^{(2)} = &E_0 \sum_{\mathfrak{l}} n_{\mathfrak{l}}, \\
H_2^{(2)} = &V_1 \sum_{\langle \mathfrak{l},\mathfrak{l}' \rangle} n_{\mathfrak{l}} n_{\mathfrak{l}'} + V_2 \sum_{\langle \mathfrak{l},\mathfrak{l}' \rangle'} n_{\mathfrak{l}} n_{\mathfrak{l}'} + V_3 \sum_{\langle \mathfrak{l},\mathfrak{l}' \rangle''} n_{\mathfrak{l}} n_{\mathfrak{l}'}, \\
H_3^{(2)} = &V_{\triangle 1} \sum_{\langle \mathfrak{l},\mathfrak{l}',\mathfrak{l}'' \rangle_1} n_{\mathfrak{l}} n_{\mathfrak{l}'} n_{\mathfrak{l}''} + V'_{\triangle 1} \sum_{\langle \mathfrak{l},\mathfrak{l}',\mathfrak{l}'' \rangle_1} n_{\mathfrak{l}} n_{\mathfrak{l}'} n_{\mathfrak{l}''} + V_{\triangle 2} \sum_{\langle \mathfrak{l},\mathfrak{l}',\mathfrak{l}'' \rangle_2} n_{\mathfrak{l}} n_{\mathfrak{l}'} n_{\mathfrak{l}''} \\
&+ V'_{\triangle 2} \sum_{\langle \mathfrak{l},\mathfrak{l}',\mathfrak{l}'' \rangle'_2} n_{\mathfrak{l}} n_{\mathfrak{l}'} n_{\mathfrak{l}''} + V''_{\triangle 2} \sum_{\langle \mathfrak{l},\mathfrak{l}',\mathfrak{l}'' \rangle''_2} n_{\mathfrak{l}} n_{\mathfrak{l}'} n_{\mathfrak{l}''} + V_{\triangle 3} \sum_{\langle \mathfrak{l},\mathfrak{l}',\mathfrak{l}'' \rangle'_3} n_{\mathfrak{l}} n_{\mathfrak{l}'} n_{\mathfrak{l}''}, \\
H_t^{(2)} = &\sum_{i,j=1}^{N}{}' t(n_{i,j-1} + n_{i,j+1})(a^+_{i+1,j} a_{i-1,j} + a_{i+1,j} a^+_{i-1,j}) \\
&+ \sum_{i,j=1}^{N}{}'' t(n_{i-1,j} + n_{i+1,j})(a^+_{i,j+1} a_{i,j-1} + a_{i,j+1} a^+_{i,j-1}),
\end{aligned}
\tag{B.19}
$$

where $V_1 = 2K_1$, $V_2 = 2K_2$, $V_3 = 2K_3$, $V_{\triangle 1} = K_{\triangle 1}$, $V'_{\triangle 1} = K'_{\triangle 1}$, $V''_{\triangle 1} = K''_{\triangle 1}$, $V_{\triangle 3} = K_{\triangle 3}$, $V_{\triangle 4} = K_{\triangle 4}$.

## C  Proof of the ground-state configuration for the localized triplons

To get the proof for the ground state of the effective model in case of localized triplons, we need to decompose the total Hamiltonian (6) into the sum of the local clusters composed of eight dimers (see Fig. 13):

$$H = \sum_{i,j=1}^{N}{}' H'_{i,j} + \sum_{i,j=1}^{N}{}'' H''_{i,j}, \tag{C.1}$$

where

$$
\begin{aligned}
H'_{i,j} =&(e_0 + h_{c1} - h)[\alpha_0(n_{i,j} + n_{i+1,j} + n_{i,j+1} + n_{i+1,j+1}) \\
&+ \alpha_1(n_{i-1,j+1} + n_{i,j-1} + n_{i+1,j+2} + n_{i+2,j})] \\
&+ V_1[\gamma_0^{(1)}(n_{i,j}n_{i+1,j+1} + n_{i,j+1}n_{i+1,j}) \\
&+ \gamma_1^{(1)}(n_{i,j}n_{i-1,j+1} + n_{i+2,j}n_{i+1,j+1} + n_{i,j-1}n_{i+1,j} + n_{i,j+1}n_{i+1,j+2})] \\
&+ V_2[\gamma_0^{(2)}(n_{i-1,j+1}n_{i+1,j} + n_{i,j+1}n_{i+2,j} + n_{i,j-1}n_{i+1,j+1} + n_{i,j}n_{i+1,j+2}) \\
&+ \gamma_1^{(2)}(n_{i-1,j+1}n_{i,j-1} + n_{i,j-1}n_{i+1,j} + n_{i+2,j}n_{i+1,j+2} + n_{i+1,j+2}n_{i-11,j+1})] \\
&+ V_3[n_{i-1,j+1}n_{i+2,j} + n_{i,j-1}n_{i+1,j+2}],
\end{aligned}
$$

$$
\begin{aligned}
H''_{i,j} =&(e_0 + h_{c1} - h)[\alpha_0(n_{i,j} + n_{i+1,j} + n_{i,j+1} + n_{i+1,j+1}) \\
&+ \alpha_1(n_{i-1,j} + n_{i+1,j-1} + n_{i,j+2} + n_{i+2,j+1})] \\
&+ V_1[\gamma_0^{(1)}(n_{i,j}n_{i+1,j+1} + n_{i,j+1}n_{i+1,j}) \\
&+ \gamma_1^{(1)}(n_{i,j}n_{i+1,j-1} + n_{i-1,j}n_{i,j+1} + n_{i,j+2}n_{i+1,j+1} + n_{i+1,j}n_{i+2,j+1})] \\
&+ V_2[\gamma_0^{(2)}(n_{i-1,j}n_{i+1,j+1} + n_{i,j}n_{i+2,j+1} + n_{i,j+1}n_{i+1,j-1} + n_{i,j+2}n_{i+1,j}) \\
&+ \gamma_1^{(2)}(n_{i-1,j}n_{i+1,j-1} + n_{i-1,j}n_{i,j+2} + n_{i,j+2}n_{i+2,j+1} + n_{i+2,j+1}n_{i+1,j-1})] \\
&+ V_3[n_{i-1,j}n_{i+2,j+1} + n_{i+1,j-1}n_{i,j+2}], \tag{C.2}
\end{aligned}
$$

$\alpha_i$ and $\gamma_i^{(l)}$ are free parameters, which we fix later depending on the region of the ground-state phase diagram. Since sites and bonds may belong to several cluster Hamiltonians (C.2), the normalization conditions are required to secure that each of them is counted only once, i.e

$$4\alpha_0 + 4\alpha_1 = 1, \ \gamma_0^{(1)} + 2\gamma_1^{(1)} = 1, \ \gamma_0^{(2)} + \gamma_1^{(2)} = 1. \tag{C.3}$$

It should be noted that all three-particle couplings in the Hamiltonian (6) as well as the correlated hopping terms are excluded preserving only pair interactions in the effective Hamiltonian (C.1). The analysis of the correlated hopping terms is considered in Sec. 4. The rigorous treatment of the three-particle couplings is still possible but it would require larger clusters and a more intricate analysis. On the other hand, weak three-particle interactions do not contribute to the low-field phases up to 1/6-plateau at all. As we show below they lift the degeneracy in the 1/4-plateau phase selecting the stripe-like phase (see Fig. 8(d)).

The idea to prove the ground state is based on the variational principle. At first we determine the allowed configurations of the cluster Hamiltonians (C.2) with the lowest eigenenergy.

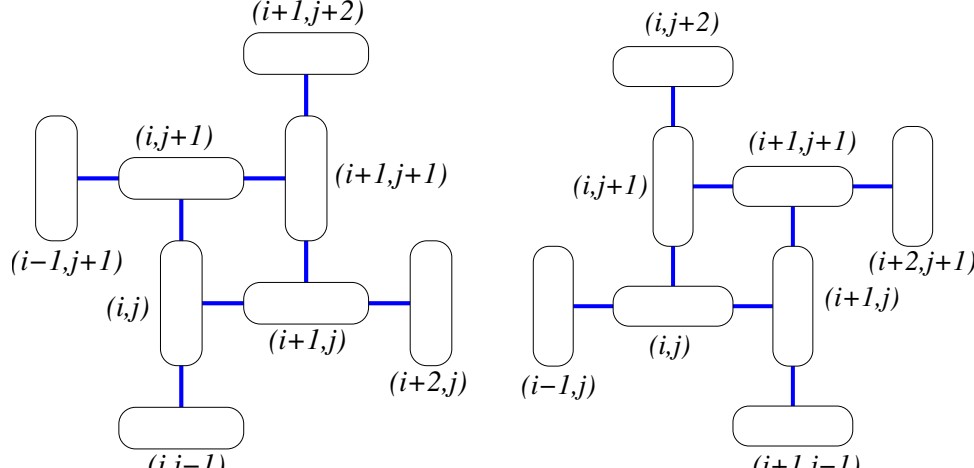

Figure 13: The local clusters of eight dimers given by Hamiltonians (C.1).

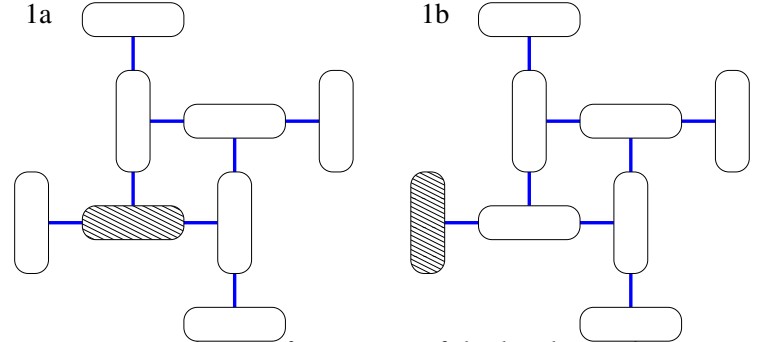

Figure 14: One-triplon configurations of the local Hamiltonians (C.1).

If we are able to construct some global state consistent with each allowed local configuration, this state will acquire the lowest eigenenergy as the ground state.

The "vacuum" state for the effective Hamiltonian as well as for each local Hamiltonian (C.2) has zero energy. All other possible topologically nonequivalent configurations with triplons are given in Figs. 14–16. Their energies can be readily calculated from Eqs. (C.2):

$$
\begin{aligned}
E_{1a} &= \alpha_0(e_0 + h_{c1} - h),\\
E_{1b} &= \alpha_1(e_0 + h_{c1} - h),\\
E_{2a} &= 2\alpha_1(e_0 + h_{c1} - h) + V_3,\\
E_{2b} &= \frac{1}{4}(e_0 + h_{c1} - h) + \gamma_0^{(2)} V_2,\\
E_{2c} &= 2\alpha_1(e_0 + h_{c1} - h) + \gamma_1^{(2)} V_2,\\
E_{2d} &= 2\alpha_0(e_0 + h_{c1} - h) + \gamma_0^{(1)} V_1,\\
E_{2e} &= \frac{1}{4}(e_0 + h_{c1} - h) + \gamma_1^{(1)} V_1,\\
E_{3a} &= (\alpha_0 + 2\alpha_1)(e_0 + h_{c1} - h) + \gamma_1^{(1)} V_1 + V_2.
\end{aligned}
\tag{C.4}
$$

For low fields the free parameters can be set equally, i.e. $\alpha_0 = \alpha_1 = 1/8$, $\gamma_0^{(1)} = \gamma_1^{(1)} = 1/3$, $\gamma_0^{(2)} = \gamma_1^{(2)} = 1/2$. It is evident that the configuration with zero triplons corresponds to the lowest energy constituting the singlet-dimer phase till $(e_0 + h_{c1} - h) > 0$. Otherwise the configurations with triplet excitations may attain lower energies. Thus, the upper field of the

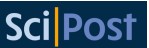

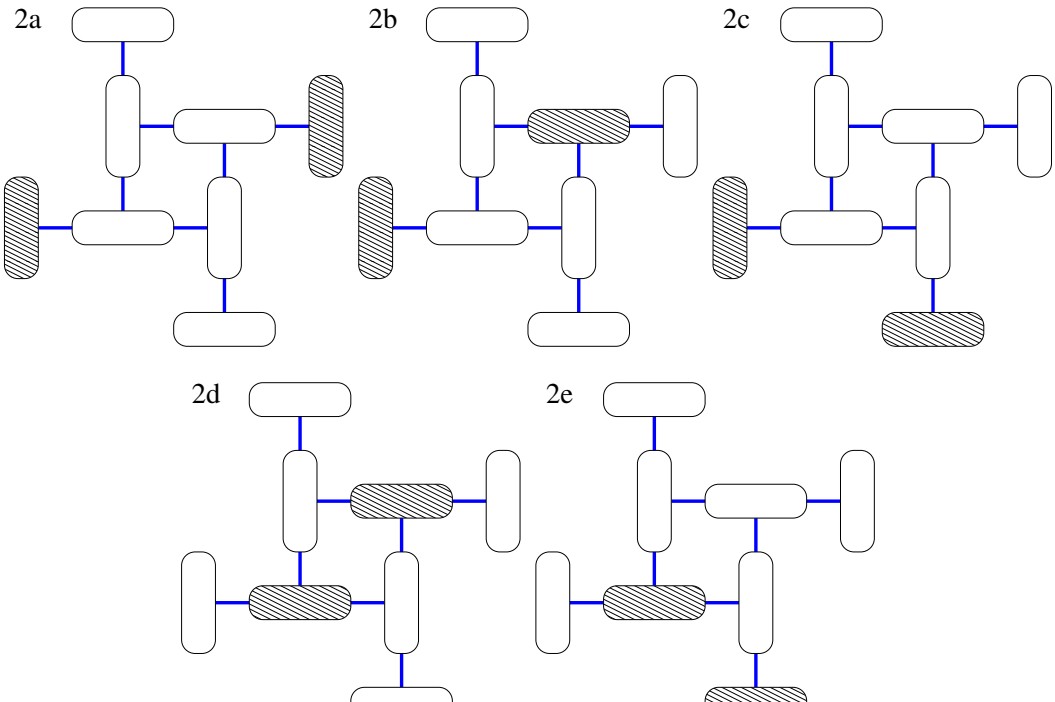

Figure 15: Two-triplon configurations of the local Hamiltonians (C.1).

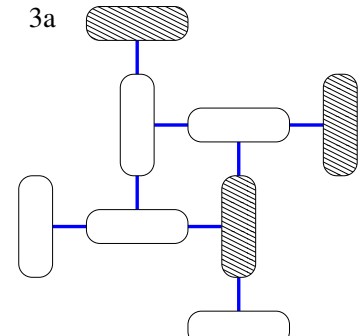

Figure 16: Three-triplon configurations of the local Hamiltonians (C.1).

singlet-dimer phase corresponding to zero plateau is

$$h_{0-1/8} = e_0 + h_{c1}. \tag{C.5}$$

It is easy to show that right above $h_{0-1/8}$ one-triplon configurations (1a) and (1b) have the lowest energy among others. It becomes evident, since all pair couplings are repulsive. In the aforementioned configurations only one of eight dimers is in the triplet state. That means that any phase made of these configurations corresponds to the 1/8-plateau phase. To get all possible ground-state configurations in this case one has to fill up the lattice with (1a) and (1b) clusters only. It can be shown by direct construction that the 1/8-plateau phase is highly degenerate, and can be built as a mixture of the vertical and horizontal ordering configurations (see Fig. 17).

Upon increasing of the magnetic field, the energy of two-triplon configuration (2a) approaches the energy of one-triplon states $E_{1a} = E_{1b}$. The condition $E_{2a} = E_{1a} = E_{1b}$, thus, defines the upper limit of 1/8 plateau phase:

$$h_{1/6-1/8} = h_{c1} + e_0 + 8V_3. \tag{C.6}$$

It is interesting to observe that the addition of another configuration (2a) increase substantially the number of possible ground state configurations and the states with any magnetization

(a) (b)

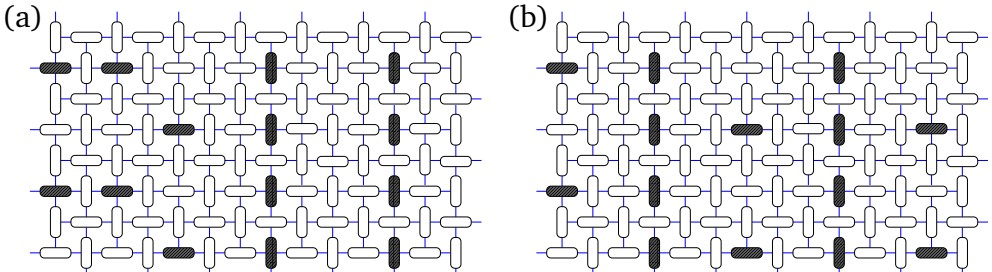

Figure 17: A few typical (disordered) configurations of triplons pertinent to the 1/8-plateau phase.

(a) (b)

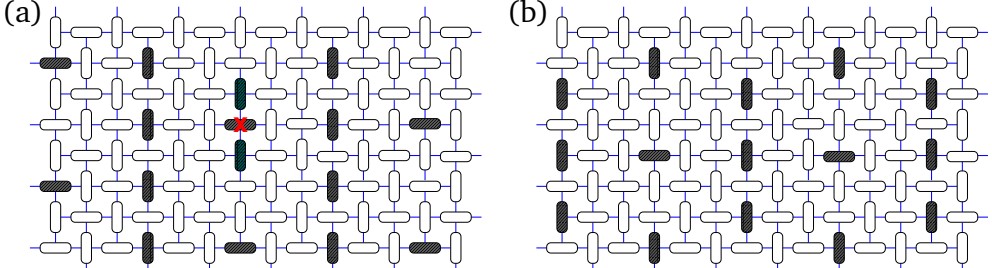

Figure 18: Intermediate phases between the 1/8- and 1/6- plateaux (left panel), and an example of the 2/15-plateau configuration (right panel).

between plateaux 1/8 and 1/6 magnetization becomes possible. For instance, we may replace a triplon state on the horizontal dimer (on the right panel of Fig. 17) to two triplons on the neighboring vertical dimers without cost of energy (see Fig. 18). Thus continuing such kind of replacement, we may achieve the configurations with any magnetization between the 1/8- and 1/6-plateaux. A particular example of 2/15-plateau is given on the right panel of Fig. 18.

To prove that 1/6-plateau phase is the ground state above $h_{1/6-1/8}$, we need to force that the cluster configurations (1a) and (2a) that creates this phase have equal energy. That is $E_{2a} = E_{1a}$. This condition leads to

$$\alpha_0 = \frac{1}{3}\left(\frac{1}{2} + \frac{V_3}{E_0}\right), \; \alpha_1 = \frac{1}{3}\left(\frac{1}{4} - \frac{V_3}{E_0}\right),$$
$$E_{2a} = E_{1a} = \frac{1}{6}(E_0 + 2V_3). \tag{C.7}$$

In order to find the upper limit of the 1/6-plateau we need to imply that the energies of configurations (2d) and (2e) are equal $E_{2d} = E_{2e}$. This fixes values of the variational parameters $\gamma_0^{(1)}$ and $\gamma_1^{(1)}$:

$$\gamma_0^{(1)} = \frac{1}{3V_1}\left(\frac{1}{12}E_0 + \frac{2}{3}V_3 + V_1\right),$$
$$\gamma_1^{(1)} = \frac{1}{3V_1}\left(-\frac{1}{6}E_0 - \frac{4}{3}V_3 + V_1\right). \tag{C.8}$$

So that the energy of this configurations is as follows

$$E_{2d} = E_{2e} = \frac{5}{18}E_0 + \frac{2}{9}V_3 + \frac{1}{3}V_1. \tag{C.9}$$

From the condition $E_{2a} = E_{2d}$ we find the transition from the 1/6- to 1/4-plateau:

$$h_{1/6-1/4} = h_{c1} + e_0 + 3V_1 - V_3. \tag{C.10}$$

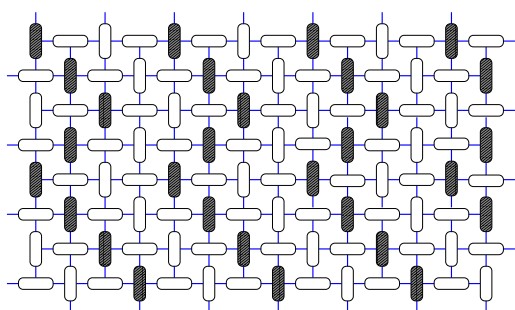

Figure 19: Zigzag-like configuration for the 1/4-plateau phase.

The stability of the 1/4-plateau is ensured by the configurations with equal energies $E_{2a} = E_{2d} = E_{2e}$. We find the following values for the variational parameters:

$$\alpha_1 = \frac{1}{8E_0}(E_0 - V_1 + 3V_3), \ \alpha_0 = \frac{1}{4} - \alpha_1,$$

$$\gamma_0^{(1)} = \frac{2}{3V_1}\left(-2\alpha_0 E_0 + \frac{1}{4}E_0 + \frac{1}{2}V_1\right),$$

$$\gamma_1^{(1)} = \frac{1}{3V_1}\left(2\alpha_0 E_0 - \frac{1}{4}E_0 + V_1\right). \tag{C.11}$$

In consequence of that, the energies of all selected configurations are equal

$$E_{2a} = E_{2d} = E_{2e} = \frac{1}{4}(E_0 + V_1 + V_3). \tag{C.12}$$

In general, the 1/4-plateau phase built from the configurations $(2a)$, $(2d)$, $(2e)$ is degenerate and may have a zigzag-like form shown in Fig. 19. However, the three-particle interaction lifts the degeneracy selecting as the ground state only the stripe phase shown in Fig. 8(d). It can be understood from Fig. 5 that $V_{\Delta 1} > 0$ increases the energy of the zigzag-like configurations, and $V'_{\Delta 1} < 0$ favors the linear ordering of three subsequent triplons.

While increasing the magnetic field, another configuration $(3a)$ reaches the energy level of the aforementioned configurations of the 1/4 plateau, i.e. $(E_{2a} = E_{2d} = E_{2e}) = E_{3a}$. It signals for the transition from 1/4- to 1/3-plateau at the critical field:

$$h_{1/4-1/3} = e_0 + h_{c1} + V_1 + 8V_2 - 3V_3. \tag{C.13}$$

There is no macroscopic degeneracy at the boundary. The ground state shows a stripe disorder, where the diagonal stripes of triplons are separated by either two or three stripes of singlet-like dimers.

The stability of the 1/3-plateau can be satisfied if $E_{3a} = E_{2d} < E_{2a}$, i.e. the 1/3-plateau is composed of two former configurations only. Therefore one can set:

$$\gamma_1^{(1)} = \frac{1}{2},$$

$$\alpha_1 = \frac{1}{3E_0}\left(\frac{1}{4}E_0 + \frac{1}{2}V_1 - V_2\right). \tag{C.14}$$

Finally, it is worthwhile to recall that all transition fields were found when disregarding the weak three-particle interactions. To get the enhanced values for the transition fields between the 1/6-, 1/4-, 1/3-plateaux, it is necessary to recalculate the energies of these states with the three-particle couplings. Such a procedure recovers the transition fields (8) presented in the main text.

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
