# Peer review of "Fractional magnetization plateaux of a spin-1/2 Heisenberg model on the Shastry-Sutherland lattice: effect of quantum XY interdimer coupling"

_SciPost Physics, doi:SciPost Phys. 12, 056 (2022)_

## Round 2 · Referee Report · Anonymous (Referee 1) · 2021-5-18

Strengths

1) Well written 2) Interesting angle of attack for a complicated problem in frustrated magnetism

Weaknesses

No severe weakness.

Report

The article entiteled "Fractional magnetization plateaux of a spin-1/2 Heisenberg model on the Shastry-Sutherland lattice: effect of quantum XY interdimer coupling" by Verkholyak and Strecka investigates the well known Shastry-Sutherland model in a field, which is to a very good level a microscopic model for describing the physics of the highly frustrated quantum magnet SrCu_2(BO_3)_2. Technically, the authors apply a perturbative treatment about the exactly solvable (non-trivial) Ising-Heisenberg model in a field, which builds on the same approach introduced by the same authors before (Ref. [47]). In this limit the ground-state phase diagram displays four phases (in the regime of interest): the singlet-dimer phase, a 1/3 plateau phase, a 1/2 plateau phase as well as a saturated phases. At the critical magnetic field between the singlet-dimer and 1/3-plateau phase, the system is macrocopically degenerate. In the current work the authors use second-order degenerate perturbation theory to derive an effective Hamiltonian in terms of interacting hardcore bosons, which includes strong two- and three-particle interactions as well as so-called correlated hopping terms (direct hopping is not present in this order, which is consistent with older studies and originates from the geometric frustration). Neglecting the correlated hopping terms, the system becomes effectively diagonal (classical) so that the phase diagram contains further plateau structures at magnetization 1/6 and 1/8. Finally, the authors study the possibility of quantum correlated plateau structures which benefit from the correlated hopping terms. Indeed, such quantum phases have been suggested by iPEPS as well as DMRG for quasi-2D cylinder structures. In my opinion the article is well written and the topic is interesting. Globally, I therefore recommend publication in SciPost. In particular, (3) it opens an interesting pathway in an existing or a new research direction, with clear potential for multipronged follow-up work. I nevertheless some points, which the authors should address to further improve their manuscript.

Requested changes

1) Page 1: In the first paragraph the authors refer to the possibility of topological triplon modes. I think the first reference in this context would be

@article{cite-key, Abstract = {SrCu2(BO3)2 is the archetypal quantum magnet with a gapped dimer-singlet ground state and triplon excitations. It serves as an excellent realization of the Shastry--Sutherland model, up to small anisotropies arising from Dzyaloshinskii--Moriya interactions. Here we demonstrate that these anisotropies, in fact, give rise to topological character in the triplon band structure. The triplons form a new kind of Dirac cone with three bands touching at a single point, a spin-1 generalization of graphene. An applied magnetic field opens band gaps resulting in topological bands with Chern numbers $\pm$2. SrCu2(BO3)2 thus provides a magnetic analogue of the integer quantum Hall effect and supports topologically protected edge modes. At a threshold value of the magnetic field set by the Dzyaloshinskii--Moriya interactions, the three triplon bands touch once again in a spin-1 Dirac cone, and lose their topological character. We predict a strong thermal Hall signature in the topological regime.}, Author = {Romh{\'a}nyi, Judit and Penc, Karlo and Ganesh, R.}, Da = {2015/04/13}, Date-Added = {2021-05-18 15:47:08 +0200}, Date-Modified = {2021-05-18 15:47:08 +0200}, Doi = {10.1038/ncomms7805}, Id = {Romh{\'a}nyi2015}, Isbn = {2041-1723}, Journal = {Nature Communications}, Number = {1}, Pages = {6805}, Title = {Hall effect of triplons in a dimerized quantum magnet}, Ty = {JOUR}, Url = {https://doi.org/10.1038/ncomms7805}, Volume = {6}, Year = {2015}, Bdsk-Url-1 = {https://doi.org/10.1038/ncomms7805}}

which should be added to the bibliography. I think one should also mention that a finite DM-interaction is necessary in order to get non-trivial triplon band structures.

2) Page 6, Eq. (6): I wondered whether also pair hopping terms are present (and potentially important) in the effective description? At least in the conventional perturbation theory about the isolated dimer limit such terms are present (at low orders).

3) Page 7: In the middle of the page the authors state that interactions with more than three particles are negligibly small. Is this obvious? And if so, can one quantity this? At least also the number of such terms with increase strongly with the order of the expansion.

4) Page 8: In Ref. [6] no "classical" 1/8 plateaux is realized, since 1/9 as well as 2/15 plateaux have lower energies. While the authors comment on the fact that in their calculation 2/15 plateaus are degenerate with their 1/8 plateaus, is it also true for 1/9 (or in fact with all other structures having a lower density?)?

5) Page 8: Related to the point before: How do the authors actually determine which kind of plateaus are realized for which densities? Do they scan over a certain number of possible unit cell sizes and densities? Or is there a smarter way to indeed find the global minimum over all densities and all unit cells for a given density?

  • validity: top
  • significance: good
  • originality: good
  • clarity: high
  • formatting: excellent
  • grammar: excellent

Author:  Taras Verkholyak  on 2021-10-05  [id 1806]

(in reply to Report 1 on 2021-05-18)
Category:
answer to question
validation or rederivation
pointer to related literature

We are grateful to the referee for the high value of our manuscript and the comments.

Response to the requested changes.

Referee says:

1) Page 1: In the first paragraph the authors refer to the possibility of topological triplon modes. I think the first reference in this context would be Judit Romhányi, Karlo Penc, R. Ganesh, Nature Communications 6, 6805 (2015) which should be added to the bibliography. I think one should also mention that a finite DM-interaction is necessary in order to get non-trivial triplon band structures.

Response:

We added the mentioned reference to the bibliography (see Ref.18) and commented the importance of the Dzyaloshinskii-Moriya interaction for the proper description of triplon band structures (at the end of the second paragraph of the Introduction).

Referee says:

2) Page 6, Eq. (6): I wondered whether also pair hopping terms are present (and potentially important) in the effective description? At least in the conventional perturbation theory about the isolated dimer limit such terms are present (at low orders).

Response:

Pair hopping terms appear in the sixth order of the usual perturbation theory (see Ref. 55) [55] S. Miyahara and K. Ueda, Phys. Rev. Lett. 82, 3701 (1999). Therefore, its amplitude supposed to be rather small and does not have any substantial effect in the model. We comment this fact on page 8, below Fig. 6. Such a picture is possible because of the specific order of the mutually-orthogonal dimers. On the other hand, any deviation from this symmetric picture, like deviation from the orthogonal positions of dimers, or anisotropic couplings, like Dzyaloshinskii-Moriya one may induce the hopping terms between nearest neighbors.

Referee says:

3) Page 7: In the middle of the page the authors state that interactions with more than three particles are negligibly small. Is this obvious? And if so, can one quantity this? At least also the number of such terms with increase strongly with the order of the expansion.

Response:

The general arguments can be as follows. In the usual perturbation theory, when we start from the limit of the non-interacting dimers, the many-particle interactions emerge in the effective models in the higher order of the perturbation theory, so that three-particle interaction is obtained in the third-order perturbation theory and so on. In our treatment we take into account the Ising part of the interdimer coupling rigorously, which would be the case if we were able to sum up all this terms in the usual perturbation scheme. Therefore, they should be of the same order as in the usual perturbation scheme. To the best of our knowledge, there is no known results for the many-triplon interactions in the effective model. And our work is the first attempt to examine the importance of such terms.

Referee says:

4) Page 8: In Ref. [6] no "classical" 1/8 plateaux is realized, since 1/9 as well as 2/15 plateaux have lower energies. While the authors comment on the fact that in their calculation 2/15 plateaus are degenerate with their 1/8 plateaus, is it also true for 1/9 (or in fact with all other structures having a lower density?)?

Response:

Yes, the effective model is macroscopically degenerate at h_{0-1/8}, the transition field from the singlet-dimer to the 1/8 plateau phase. Due to the construction, the ground-state in this case can be described by the triplon gas with the hard-core condition when all possible pair couplings between them is excluded. In this case any magnetization between zero and 1/8 is possible. It is also plausible that the next order of the perturbation theory will induce the further neighbor interactions between triplons, and therefore, the phases with even lower fractional number, say 1/9, may be stabilized as well. The same is true also at the boundary between 1/8 and 1/6 plateaus, where the ground state is also macroscopically degenerate. In Appendix C, it is shown how 2/15 plateau can be constructed in this case. And it might also be stabilized in the third-order of the perturbation theory.

Indeed, in Ref. 6 no 1/8 plateau has been found, but it was revealed in the latter paper [7], as some others. This example shows how hard is the study of the magnetization plateaus. Our research build the hierarchy of the magnetization plateaus, i.e. the Ising interdimer coupling is responsible for the wide 1/3, 1/2 plateaus, and smaller plateaus of 1/8, 1/6, 1/4 come out within the second-order perturbation theory. Additional plateaus may appear between zero and 1/8, as well as between 1/8 and 1/6, but the sizes of these plateaus might be rather tiny.

Referee says:

5) Page 8: Related to the point before: How do the authors actually determine which kind of plateaus are realized for which densities? Do they scan over a certain number of possible unit cell sizes and densities? Or is there a smarter way to indeed find the global minimum over all densities and all unit cells for a given density?

Response:

We added Appendix C to the manuscript where it is described how to prove the ground-state phase diagram for the case of the localized triplons when the correlated hopping terms are excluded.

---

## Round 2 · Referee Report · Anonymous (Referee 2) · 2021-5-19

Strengths

1) methodology is explained very well 2) broader and rather complete review on this topic is provided 3) details of the calculations are explicitly included 4) illustrative figures included

Weaknesses

no serious weaknesses spotted

Report

The focus of this study is the non-trivial sequence of magnetization plateaus in the frustrated Shastry-Sutherland model, that is closely realized in the compound SrCu2(BO3)2. To that end, the Authors use a novel perturbative approach around the Ising-Heisenberg limit, treating the interdimer XY coupling as a perturbation around a highly-degenerate critical field, whose unperturbed ground state manifold had been characterised exactly previously. This is quite a non-trivial and elaborate calculation which delivers an effective low-energy Hamiltonian in terms of interacting triplons, whose minimization at various limits unravels a series of plateaus. The paper is very well written, with a rather complete review of the wide literature on this topic, and with relevant details provided in appendices. The results are very interesting, and the methodology is quite novel and promising for further applications in related models. I therefore recommend the paper for publication.

Requested changes

Minor comments: 1) there is a typo in the first term of Eq. 1: SlSl whould be replaced with SlSm. 2) I would like to see some more details on how the Authors minimized the interatcing triplon Hamiltonian to produce the various plateau phases. 3) A comment on what happens between the plateau phases (i.e., the compressible regions of the magnetization process in the regime of interest) would be useful for readers, given in particular that hopping terms are drastically quenched, and perhaps any relevance of the role of anisotropic interactions in this respect.

  • validity: top
  • significance: high
  • originality: high
  • clarity: high
  • formatting: excellent
  • grammar: good

Author:  Taras Verkholyak  on 2021-10-05  [id 1807]

(in reply to Report 2 on 2021-05-19)
Category:
answer to question
correction
validation or rederivation

We are grateful to the referee for the high value of our manuscript and the comments.

Responses to the minor comments:

Referee says:

1) there is a typo in the first term of Eq. 1: SlSl would be replaced with SlSm.

Response:

Thank you very much for pointing out on this error. We corrected it in the revised version.

Referee says:

2) I would like to see some more details on how the Authors minimized the interacting triplon Hamiltonian to produce the various plateau phases.

Response:

We explored the method how to find the plateau phases in Appendix C.

Referee says:

3) A comment on what happens between the plateau phases (i.e., the compressible regions of the magnetization process in the regime of interest) would be useful for readers, given in particular that hopping terms are drastically quenched, and perhaps any relevance of the role of anisotropic interactions in this respect.

Response:

In Appendix C we added the discussion about the phase diagram and explore the degeneracy at the transition fields between different plateau phases. We show that the boundary between zero and 1/8 plateau, as well as between 1/8 and 1/6 plateau is highly degenerate, while the boundaries between higher plateaus are not.

---

## Round 2 · Referee Report · Anonymous (Referee 3) · 2021-6-3

Strengths

1) systematic methodology 2) topical subject

Weaknesses

1) unclear presentation 2) unclear figures

Report

The authors perform a perturbative treatment of the "Ising-Heisenberg" model on the orthogonal dimer lattice with a view to understanding the cascade of magnetisation plateaux observed numerically in the Shastry-Sutherland model(SSM) and experimentally in SrCu_2(BO_3)_2 (SCBO). The details of the analysis appear solid, but the presentation suffers from a number of shortcomings that make it difficult for the reader to follow the logic and impact of the results. When the authors improve these issues the manuscript may become suitable for publication

The foundational concept of the work, a perturbation theory about an exactly known state that captures some key properties of the target model, is a good one. However, the perturbative parameter is J_xy'/J' and this fact becomes completely lost in the manuscript. If the validity of the analysis is to be tested, J_xy' must be discussed, but it appears in none of the figures (presumably it has been set to 1 in Figs. 6 and 8, but this is not stated). If the authors wish to claim that their analysis is valid even at J_xy' = 1 because of the small coefficients appearing in Fig. 6, this feature of the treatment is lost completely. It is clear from the lengthy equations that the correlated hopping terms appear at the same order as the two-body density-density terms, not the three-body ones, and thus the logic of ignoring them compared to the latter (in fixing stripe phases in the discussion) is also unclear. Given this situation, the authors may wish to reconsider their title, because no investigation of the effects of J_xy' (by changing its value) is actually presented.

In Sec. 1, "spin-electron" coupling has little meaning out of the context of the paper being referenced (the spin is carried by electrons) and more words are needed to explain this.

In Sec. 2, the physics behind the exclusion of the six-site cluster, and not a four-site one (or any other one), needs to be explained for the self-contained nature of the manuscript, and cannot simply be an arbitrary assertion. In Fig. 2 the stripe-like nature and 1/3 character are lost because of the red triplons; the authors should devise a way to show both features in the same figure.

In Sec. 3, this referee has no criticism of the mechanical aspects of the calculation (most of which are relegated to App. B), only about the presentation of the results. See above for the perturbative nature of the analysis. The main text alone contains an impossible jump to the results of Eq. (6), and its self-contained nature would be aided very considerably by (i) a summary of the fact that t and V are systematic functions of J', J_xy' and h and (ii) a better discussion of how the J_xy' terms generate the physics of the terms shown in Figs. 3-5.

The correlated hopping term is a particular problem in this regard. In this referee's understanding, the physics of assisted hopping in the SSM is that the intermediate dimer should be in a triplet state and this acts to remove the exact frustration, generating hopping terms at 2nd order instead of 6th. It is not at all clear from the equations or figures (5 or 9) where the intermediate triplet appears in the formalism, and certainly not in the discussion. If this is a different result, i.e. the present analysis involves a different type of correlated hopping, then the authors should take care to present a clear explanation of the situation.

The discussion on P7-10 is extremely difficult to read without a ground-state phase diagram. Such a diagram appears in Fig. 8, but it is not clear to which part of the discussion it refers (does it contain the t terms or not) ? In its current form, it is not clear if the authors are even describing a result, or only a series of speculations and cartoons. If the authors wish to make this section make sense, they should provide a phase diagram, or line for one illustrative J'/J, and accompany it by a graph of \Delta E(H), the energy of the ground state below the lowest competing state. Further issues in this section are (i) why is it a surprise that correlated hopping exceeds V_1 and V_2 at low J'/J ? It should be possible to read the reasons from the equations. (ii) The discussion of the 2/15 state is completely confusing: it appears in no figures (cartoons or phase diagrams) and here it is written "quite plausible to conjecture [that it] may eventually emerge ..." yet in Sec. 5 one may read "it has been found that the 2/15-plateau phase coexists." It is necessary to explain the real situation. (iii) Given that the 3-body V terms have different signs (Fig. 6(b)), is it possible to be more explicit about how which ones stabilise which stripe states ? (iv) It is not at all clear (from the text or caption) whether the correlated hopping terms are included in Fig. 8: the implication of the text is that they are simply dropped because of the cartoon configurations shown in Fig. 7. If more has been done than this, the authors would do themselves a favour to explain it. The caption of Fig. 8 is also missing an explanation of the inset.

The final two paragraphs of the section have significant logic problems. First, the theory is for the SSM and the results are being extracted from data on SCBO, which has 3D coupling and DM interactions. Second, the authors just stated that the relative width of their 1/8 plateau does not match experiment at all. Third, the analysis of Ref. [52] uses exact diagonalisation on a very small cluster that is known from much more recent and detailed work to provide a very poor account of the thermodynamics. Fourth, the authors take experimental values accurate to 1 T and report theoretical values accurate to 0.01 T. None of this makes any sense without a meaningful inclusion of error bars and explanation of which level of effective fit is being attempted. Finally, the last paragraph is not about the "validity" of the perturbation theory (see below for "small enough"). It is about the applicability of a T = 0 perturbation theory (whose validity is a function of J_xy' -- above) at finite temperatures. The suggested figure of \Delta E would make the applicability clear. The authors should also specify the value of h for the gap and V values they provide in the text.

In Sec. 4, the presentation is extremely confusing, to the extent that this referee could not understand what the authors wish to say. The analysis appears to be for propagating bosonic two-triplon objects with a well defined k-space dispersion [Eq. (11)]. First this is called a "wave" and then suddenly it is "stripe-like" but still delocalised. Are the bosons now assumed to have crystallised ? The cited Fig. 2 contains no explanation. How many of them are present per unit volume if they border the 1/8 plateau ? If the picture is a "sliding bound-state density wave" then this needs a picture and much better explanation than repeating the term "stripe" that was also used in Fig. 7.

In Sec. 5 there are problems related to the unclear issues listed elsewhere. In the end, can the authors make some definite statements a but what we have learned ? If the primary achievement is to match the numerics, it seems that nothing has been gained. Do the authors want to argue that their results are more accurate than the numerics, given that all methods have their limitations and the theory treats an infinite system ? The 2/15 plateau was missed: if it were to appear at higher order, as "speculated," could one claim real triumph ? Have we gained any predictive power, or ease of covering large parts of parameter space without expensive numerics, or a comprehensive means of fitting every aspect of experiment (see above) ?

The references have a number of problems. [2] certainly does not review "numerical methods like CORE, pCUTs, tensor network iPEPS ..." [4] and [46] are chapters of the same book and the name of the book, not just the chapter, needs to be included. [17,18] exclude the original reference to topological triplons, Nat. Commun. 6, 6805 (2015). [22] is not about QMC. [24] is a study of SCBO, not the SSM (and it observes crossovers, not transitions -- stated wrongly in the abstract but correctly in the summary of that reference).

Some confusing sentences that damage the understanding of the paper can be ascribed clearly to subtleties of English. (i) In line 2 of Sec. I the SSM is "A modern playground ..." (it is certainly not the only one). (ii) The authors need to review their use of "whereas:" this word implies a contrast to the other half of the sentence and it cannot be interchanged with "where" or "whereby." (iii) The authors need to use more words to specify "low" fields (Sec. 1) and "low enough" and "sufficiently low" in Secs. 3 and 4: all of these terms imply H --> 0 (T --> 0) and this is not at all what is meant (the physics is perfectly well understood in this regime, and the special conditions for the bound-triplon phase are not in it).

In summary, the work presents a systematic perturbative analysis of potential value to the understanding of the SSM, but the interpretation of its results is currently too confused and confusing to merit publication.
  • validity: good
  • significance: ok
  • originality: good
  • clarity: poor
  • formatting: acceptable
  • grammar: reasonable

Author:  Taras Verkholyak  on 2021-10-05  [id 1808]

(in reply to Report 3 on 2021-06-03)
Category:
answer to question
reply to objection
correction
validation or rederivation

We are grateful to the referee for carefully reading our manuscript and for the remarks.

Responses to the remarks and requested changes:

Referee says:
The details of the analysis appear solid, but the presentation suffers from a number of shortcomings that make it difficult for the reader to follow the logic and impact of the results. When the authors improve these issues the manuscript may become suitable for publication

Response:
We revised the manuscript to improve the presentation.

Referee says:
The foundational concept of the work, a perturbation theory about an exactly known state that captures some key properties of the target model, is a good one.
However, the perturbative parameter is J_xy'/J' and this fact becomes completely lost in the manuscript. If the validity of the analysis is to be tested, J_xy' must be discussed, but it appears in none of the figures (presumably it has been set to 1 in Figs. 6 and 8, but this is not stated). If the authors wish to claim that their analysis is valid even at J_xy' = 1 because of the small coefficients appearing in Fig. 6, this feature of the treatment is lost completely.
...
Given this situation, the authors may wish to reconsider their title, because no investigation of the effects of J_xy' (by changing its value) is actually presented.

Response:
We should stress that the idea of our work was not study the dependence on the parameter J'_xy, but to analyze the origin of the magnetization plateaus in the Shastry-Sutherland model. Previously, we showed that the broad 1/3 and 1/2 plateaus emerge already in the Ising-Heisenberg model on the Shastry-Sutherland lattice [51], where the quantum part of the interdimer coupling is disregarded. It served as evidence that the XY part of the interdimer coupling might be responsible for smaller fractional plateaus. And our current study supports this point of view. We showed that the second-order perturbation theory is able to predict additional 1/8, 1/6, and 1/4 plateaus, while higher-order perturbation terms may describe even smaller plateaus between zero and 1/8, as well as between 1/8 and 1/6 magnetization. We changed the conclusions to stress this point explicitly.
The perturbative term is defined in Eq.(5). It has a form of the XY part of the interdimer coupling with the constant J'. We did introduce the separate notation for the constant of the XY part in Appendix B only in order to highlight the order of the perturbative terms. Since the original model which we consider is isotropic, we take J'_z = J'_xy in all final expressions.
What regards the validity of the perturbation theory, it is well known that the convergence of the perturbation theory is hard to prove rigorously. The present case is not an exception. On the other hand, we tested the method on the orthogonal-dimer chain [60], where a good agreement with the numerical methods has been found. In addition, our current results show a good agreement with the available numerical data on the Shastry-Sutherland model in a wide range of J'/J as well.

Referee says:
It is clear from the lengthy equations that the correlated hopping terms appear at the same order as the two-body density-density terms, not the three-body ones, and thus the logic of ignoring them compared to the latter (in fixing stripe phases in the discussion) is also unclear.

Response:
Indeed, the correlated hopping term is essential and cannot be generally ignored, although it competes with the strong repulsion between neighboring triplons. We showed that it is important for the low-density phase, which we discussed separately in Sec.4. We rewrote Sec.4 to improve its clarity. In case of more dense phases starting from the 1/8 plateau phase, the kinetic terms are obscured by the extensive hard-core conditions. The complete analysis of the correlated hopping term in the hard-core gas is a complex problem which deserves the separate study.

Referee says:
In Sec. 1, "spin-electron" coupling has little meaning out of the context of the paper being referenced (the spin is carried by electrons) and more words are needed to explain this.

Response:
We have changed the mentioned sentence in the Introduction (see the text on page 3):
"...the possible exchange coupling between localized Ising spins and the spins of conducting electrons
in the metallic rare-earth tetraborides may be essential for a description of their magnetic properties [49,50]."

Referee says:
In Sec. 2, the physics behind the exclusion of the six-site cluster, and not a four-site one (or any other one), needs to be explained for the self-contained nature of the manuscript, and cannot simply be an arbitrary assertion.

Response:
We extended the explanation regarding the hard-core condition in Sec. 2 (see the text on page 6).

Referee says:
In Fig. 2 the stripe-like nature and 1/3 character are lost because of the red triplons; the authors should devise a way to show both features in the same figure.

Response:
We changed Fig. 2 (Fig.3 in the revised version) removing the red color in the dimer, since it might be confusing for readers. Instead we indicate the hard-core condition by shading the corresponding area with light-blue color.

Referee says:
In Sec. 3, this referee has no criticism of the mechanical aspects of the calculation (most of which are relegated to App. B), only about the presentation of the results. See above for the perturbative nature of the analysis. The main text alone contains an impossible jump to the results of Eq. (6), and its self-contained nature would be aided very considerably by (i) a summary of the fact that t and V are systematic functions of J', J_xy' and h and (ii) a better discussion of how the J_xy' terms generate the physics of the terms shown in Figs. 3-5.

Response:
We added the comment about the parameters of the effective model after Eq. (6). We left all the technical details of the calculation to Appendix B, including an explicit form of the effective interaction parameters, in order not to overload the main text.

Referee says:
The correlated hopping term is a particular problem in this regard. In this referee's understanding, the physics of assisted hopping in the SSM is that the intermediate dimer should be in a triplet state and this acts to remove the exact frustration, generating hopping terms at 2nd order instead of 6th. It is not at all clear from the equations or figures (5 or 9) where the intermediate triplet appears in the formalism, and certainly not in the discussion. If this is a different result, i.e. the present analysis involves a different type of correlated hopping, then the authors should take care to present a clear explanation of the situation.

Response:
In Appendix B we added Eqs. (33), (34) and a new Fig.12 as well as the discussion how the correlated hopping terms appear in the perturbation theory.

Referee says:
The discussion on P7-10 is extremely difficult to read without a ground-state phase diagram. Such a diagram appears in Fig. 8, but it is not clear to which part of the discussion it refers (does it contain the t terms or not) ? In its current form, it is not clear if the authors are even describing a result, or only a series of speculations and cartoons. If the authors wish to make this section make sense, they should provide a phase diagram, or line for one illustrative J'/J, and accompany it by a graph of \Delta E(H), the energy of the ground state below the lowest competing state. Further issues in this section are (i) why is it a surprise that correlated hopping exceeds V_1 and V_2 at low J'/J ? It should be possible to read the reasons from the equations. (ii) The discussion of the 2/15 state is completely confusing: it appears in no figures (cartoons or phase diagrams) and here it is written "quite plausible to conjecture [that it] may eventually emerge ..." yet in Sec. 5 one may read "it has been found that the 2/15-plateau phase coexists." It is necessary to explain the real situation. (iii) Given that the 3-body V terms have different signs (Fig. 6(b)), is it possible to be more explicit about how which ones stabilise which stripe states ? (iv) It is not at all clear (from the text or caption) whether the correlated hopping terms are included in Fig. 8: the implication of the text is that they are simply dropped because of the cartoon configurations shown in Fig. 7. If more has been done than this, the authors would do themselves a favour to explain it. The caption of Fig. 8 is also missing an explanation of the inset.

Response:
We changed the discussion in Sec.3 (see pages 8-12). We stressed that we examine only the case of localized triplon there, leaving the analysis of the correlating hopping terms to Sec. 4. We also extended the discussion on the ground-state phases and on the possible macroscopic degeneracy at the transition fields between them. We added a new Appendix C, where we provided the proof of the ground-state phase diagram when assuming the localized triplon only.

Response to other remarks:
(i) the correlated hopping term is absent in the ordinary second-order perturbation theory and appears in the higher-order terms only. Therefore, it was not evident for us that its value would be of the same order that the pair interaction between triplons.
(ii) In Appendix C we added the analysis of the degenerate states at the transition fields between different phases. In particular, we discuss the possibility to obtain the 2/15 plateau phase at the boundary between 1/8 and 1/6 plateau phases. We also extended the discussion on this point on p.9-10.
(iii) We added the detailed explanation to new Appendix C.
(iv) We changed the caption to Fig. 7 (Fig. 8 in the revised version) and added the explanation which result corresponds to the case where the correlated hopping terms have been taking into account. We also added description of the inset into the caption.

Referee says:
The final two paragraphs of the section have significant logic problems. First, the theory is for the SSM and the results are being extracted from data on SCBO, which has 3D coupling and DM interactions. Second, the authors just stated that the relative width of their 1/8 plateau does not match experiment at all. Third, the analysis of Ref. [52] uses exact diagonalisation on a very small cluster that is known from much more recent and detailed work to provide a very poor account of the thermodynamics. Fourth, the authors take experimental values accurate to 1 T and report theoretical values accurate to 0.01 T. None of this makes any sense without a meaningful inclusion of error bars and explanation of which level of effective fit is being attempted. Finally, the last paragraph is not about the "validity" of the perturbation theory (see below for "small enough"). It is about the applicability of a T = 0 perturbation theory (whose validity is a function of J_xy' -- above) at finite temperatures. The suggested figure of \Delta E would make the applicability clear. The authors should also specify the value of h for the gap and V values they provide in the text.

Response:
We agree that we cannot determine the microscopic parameters for SrCu2(BO3)2 precisely, since the complete model for this compound might be more complex including the weak Dzyaloshinskii-Moriya and interlayer couplings. The idea was to sum up the possible application to SrCu2(BO3)2. We correccted the last paragraph of Sec. 3 accordingly. We also remove there the decimal digits from the theoretical values.

We removed the last paragraph of Sec. 3 (from previous version) about the applicability of the effective model to the low-temperature thermodynamics, since such an analysis should be more extensive.

Referee says:
In Sec. 4, the presentation is extremely confusing, to the extent that this referee could not understand what the authors wish to say. The analysis appears to be for propagating bosonic two-triplon objects with a well defined k-space dispersion [Eq. (11)]. First this is called a "wave" and then suddenly it is "stripe-like" but still delocalised. Are the bosons now assumed to have crystallised ? The cited Fig. 2 contains no explanation. How many of them are present per unit volume if they border the 1/8 plateau ? If the picture is a "sliding bound-state density wave" then this needs a picture and much better explanation than repeating the term "stripe" that was also used in Fig. 7.

Response:
We modified Sec.4 in order to improve clarity of our calculations. Now we start from finding the eigenstate and the eigenenergy of a single bound triplon state, and identified the critical field when the state of bound triplons appears. This is a rigorous result for the effective Hamiltonian (6). The second part of Sec.4 regards the restricted model of bound triplons as hard-core bosons where we discuss the possible origin of the quantum phase of bound triplons.

Referee says:
In Sec. 5 there are problems related to the unclear issues listed elsewhere. In the end, can the authors make some definite statements a but what we have learned ? If the primary achievement is to match the numerics, it seems that nothing has been gained. Do the authors want to argue that their results are more accurate than the numerics, given that all methods have their limitations and the theory treats an infinite system ? The 2/15 plateau was missed: if it were to appear at higher order, as "speculated," could one claim real triumph ? Have we gained any predictive power, or ease of covering large parts of parameter space without expensive numerics, or a comprehensive means of fitting every aspect of experiment (see above) ?

Response:
We modified Sec.5 to emphasize the importance and novelty of our results. A good agreement with the numerics serves as evidence that the method is able to reproduce the reliable results in a wide interval of J'/J. The main outcome is the effective model that is able to provide the consistent picture for the magnetization plateaux in the Shastry-Sutherland model and clarifies their origin which still remains in spite of considerable efforts controversial and highly debated issue yet. The distinctive feature of the effective model is the absence of the frustration. That makes it available for the quantum Monte Carlo method, and thus the quantum phase as well as the low-temperature properties is possible to examine for sufficiently large systems.

Referee says:
The references have a number of problems. [2] certainly does not review "numerical methods like CORE, pCUTs, tensor network iPEPS ..." [4] and [46] are chapters of the same book and the name of the book, not just the chapter, needs to be included. [17,18] exclude the original reference to topological triplons, Nat. Commun. 6, 6805 (2015). [22] is not about QMC. [24] is a study of SCBO, not the SSM (and it observes crossovers, not transitions -- stated wrongly in the abstract but correctly in the summary of that reference).

Response:
We changed the text according to the referee remarks.

Referee says:
Some confusing sentences that damage the understanding of the paper can be ascribed clearly to subtleties of English. (i) In line 2 of Sec. I the SSM is "A modern playground ..." (it is certainly not the only one). (ii) The authors need to review their use of "whereas:" this word implies a contrast to the other half of the sentence and it cannot be interchanged with "where" or "whereby." (iii) The authors need to use more words to specify "low" fields (Sec. 1) and "low enough" and "sufficiently low" in Secs. 3 and 4: all of these terms imply H --> 0 (T --> 0) and this is not at all what is meant (the physics is perfectly well understood in this regime, and the special conditions for the bound-triplon phase are not in it).

Response:
We changed the text according to the referee remarks.

---

## Round 2 · Referee Report · Anonymous (Referee 5) · 2021-6-7

Strengths

1- good introduction 2-systematic approach around a not so common starting point 3-essentially analytic

Weaknesses

1-unclear and uncertain conclusions
2-no take-home message beyond the known numerical results
3-unclear degree of accuracy: What is lacking and how large
are these effects?

Report

The authors consider the ground state phases of the Shastry-Sutherland
lattice in a finite magnetic field. In particular, they study the
fractional magnetization plateaus.
The chosen approach consists in standard second order perturbation
theory around the limit of a particular model: the intradimer-Heisenberg-
interdimer-Ising model. This starting point allows for an
exact enumeration of eigen states and eigen energies. The perturbation
enables the authors to set up an effective model with interactions
over some distance. These mostly repulsive interactions help to
explain the various magnetization plateaus.

This theoretical work is a solid computation of various
two- and three-body interactions of an effective model.
It is nice to see that this kind of reasoning is essentially
in-line with the numerical results of other approaches.
Yet, one has to wonder what can be learnt or understood that
is not accessible by other techniques or that was not known
before. The degree of novelty and originality of the present manuscript
is a bit limited. All in all, the manuscript rather represents
a progress report rather than a paper with definite conclusions.
An additional drawback consists in the fact that certain processes
are not included for the sake of clarity/feasibility. The reader
is left with a degree of uncertainty whether the left-out terms
or higher order terms will improve or deteriorate the agreement
obtained so far.

In addition, it must be seen that this model has been studied
intensively by a plethora of methods for over twenty years by now,
not counting the years before 2000 when the model was a theoretical
toy model only.

So I tend not to recommend its publication in SciPost Physics
in view of the high standards of this journal.

Requested changes

1-I would prefer if the complete set of second order terms were computed
and discussed.
2-To me, the presented analytical calculations are just a first step and would
need to be supplemented with an unbiased analysis of the effective model
obtained.

  • validity: good
  • significance: ok
  • originality: ok
  • clarity: ok
  • formatting: good
  • grammar: excellent

Author:  Taras Verkholyak  on 2021-10-05  [id 1810]

(in reply to Report 5 on 2021-06-07)
Category:
remark
answer to question
reply to objection
suggestion for further work

We are grateful to the referee for carefully reading our manuscript and for the remarks.

Responses to the remarks and requested changes:

Referee says:
This theoretical work is a solid computation of various two- and three-body interactions of an effective model. It is nice to see that this kind of reasoning is essentially in-line with the numerical results of other approaches. Yet, one has to wonder what can be learnt or understood that is not accessible by other techniques or that was not known before. The degree of novelty and originality of the present manuscript is a bit limited. All in all, the manuscript rather represents a progress report rather than a paper with definite conclusions. An additional drawback consists in the fact that certain processes are not included for the sake of clarity/feasibility. The reader is left with a degree of uncertainty whether the left-out terms or higher order terms will improve or deteriorate the agreement obtained so far.

Response:
We do not agree that our manuscript lacks novelty and originality. In our work we used the idea to separate the interdimer correlations in the Shastry-Sutherland model. The correlations caused by the "classical" Ising part of the coupling are responsible for the wide 1/3 plateau (what was know from the preceding paper [51]). Here we showed explicitly that the correlations stemmed from the XY part of the interdimer coupling are able to reproduce the series of smaller plateaux of 1/8, 1/6, and 1/4 of the saturation magnetization. Such a consistent description of the fractional plateau sequence has not been achieved before.

We had no goal to achieve a perfect agreement with the available numerical results. But a good agreement with them justifies a fast convergence of our perturbation expansion. The effective model that we obtained contains an essential correlated hopping term, which requires an extended study. In the current manuscript we just did the basic steps towards the understanding of the origin of the possible quantum phase in the Shastry-Sutherland model and SrCu2(BO3)2. The effective model including the correlated hopping term is amenable to the quantum Monte Carlo simulations, which can be done for the much larger system sizes in comparison with that used for the original Shastry-Sutherland model.

Referee says:
In addition, it must be seen that this model has been studied
intensively by a plethora of methods for over twenty years by now,
not counting the years before 2000 when the model was a theoretical
toy model only.

Response:
Indeed, the model has been studied a lot for many years. But it remains under active study until now, since is rather complex and there are still a lot of open problems left - controversial debate about the size and nature of intermediate plateaux remains unresolved issue to date.

REQUESTED CHANGES:

Referee says:
1-I would prefer if the complete set of second order terms were computed
and discussed.

Response:
Our results are in a good agreement with the available numerical methods. We demonstrated that already effective three-particle interactions are much smaller than the pair couplings and hence, they do not crucially alter the magnetic behavior. The same should be inferred for all many-particle effective interactions of higher order. The calculation of all second order terms will complicate the effective model and would not lead to any considerable improvement with what we have now. Moreover, such a calculation requires much more time and effort.

Referee says:
2-To me, the presented analytical calculations are just a first step and would
need to be supplemented with an unbiased analysis of the effective model
obtained.

Response:
Surely, the unbiased analysis of the effective model is desirable. The main focus is on the obtaining the effective model and its simple analysis. The extensive study deserves a separate studies, which might include not only analytical but also numerical calculations based, e.g. on quantum Monte Carlo simulations, etc.

---

## Round 2 · Referee Report · Anonymous (Referee 4) · 2021-6-7

Strengths

1 New route to addressing aspects of Shastry-Sutherland model phase diagram - a classic problem in quantum magnetism - that captures multiple plateau phases semiclassically. 2. Good introduction, referencing and description of Heff and semi-classical phases and associated figures.

Weaknesses

1 Lack of clarity in correlated hopping section.

Report

The problem of determining the phases of the Shastry-Sutherland model in a field has received much attention over the years. This paper is a new approach to an aspect of the problem: perturbation theory about an exactly known macroscopically degenerate phase boundary proximate to the zero field phase and the 1/3 magnetization plateau. This is in contrast to direct numerics on the full model or J'/J perturbation theory. In particular the paper does the following: - The starting point is the set of known ground states of the J Heisenberg, J' Ising model on the Shastry-Sutherland lattice in a magnetic field along the Ising direction. The transverse J' terms (and deviation from the critical field) are included as a perturbation to obtain the terms out to second order in ordinary Rayleigh-Schrodinger perturbation theory (described in detail in Appendix B with terms enumerated in Eq. (6)) where the zeroth order states are the macroscopically degenerate set of states at the boundary between the (zero field) singlet and (1/3 plateau) stripe states. - Taking only the classical terms in the effective Hamiltonian the ground states can be found essentially exactly resulting in the stabilization of 1/8, 1/6 and 1/4 plateaux. The 1/8 ground states have a residual macroscopic degeneracy. The authors show the perturbative effective couplings comment on the Hamiltonian terms that are important for selecting the 1/4 plateau over the other phases. The classical phase boundaries obtained in this way are in reasonable agreement with numerical studies. The perturbation theory and approximate mapping to a classical problem help to clarify why certain states are preferred over others. - The "quantum" terms - two-triplon bound states - are discussed in an approximate way. The authors focus on the states most susceptible to resonances. The authors claim that triplon pairs in such states are restricted to move in 1D - or, loosely speaking, "fractonic". Then focussing on the single particle problem on the dual lattice the authors estimate the binding energy of triplon pairs arguing that the two-triplon bound state is stable within a narrow field range that they estimate. - Comparisons are made to SrCu2(BO3)2 and the authors implicitly point out that their perturbation theory is particularly well-suited to the tetrahalide materials on this lattice.

The paper is apparently correct and I find it to be an interesting contribution to the literature on this problem complementary to other approaches. The method can be further explored to address other interesting aspects of the Shastry-Sutherland model and related materials. The introduction is clear and well-referenced and the details of calculations are mainly well-explained, the figures are mainly good. There are, however, some significant issues of presentation that should be revisited by the authors before publication.

Requested changes

1 The phase diagram in figure 8 is the central result but the abundance of detail obscures the paper's message. So, it would be good to open the paper with two schematic phase diagrams for fix J'/J and varying field: one for Ising-Heisenberg model, one for the effective Hamiltonian as a guide to the reader. In the latter schematic, please spell out that the hopping terms are not treated exactly and perhaps include the critical fields notation in (8). Since SrCu2(BO3)2 is frequently mentioned, the authors may consider including a similar rough field phase diagram for this material. Equations or at least detailed description of how to obtain the phase boundaries should be referenced from the caption.

2 If I understand correctly the 1/8 state in Fig. 7 has degenerate row and column shifts and Fig 7(b) has the middle row shifted from (a). Maybe highlight this feature in the caption. Clarify the sentence "On the other hand, the correlated hopping...": in what way is the correlated hopping suppressed? Relatedly, the fractional plateau phases are stated to be exact eigenstates of the full effective Hamiltonian so it should be possible to quote the energies including H_t, but these couplings are absent in eq. (8). Do they drop out somehow or are they actually not included?

3 Section 4 could be much clearer. The fact that this is an approximate treatment should be clarified. The precise form of the phase boundaries that go into Fig 8 should be spelt out. The restricted mobility of the quasiparticles is a crucial point but it is not evident to me from the Fig 9. nor the surrounding discussion how this constraint arises. The projectors (9) could perhaps be relegated to an appendix. In Eq. (10) presumably the P projectors should have labels. "wave" is confusing terminology - perhaps change to just "state" or "pi state". "At this critical field...should evolve": can this sentence be removed? And "evolve" in the conclusions should perhaps be "arise"? The authors seem to be saying that the two-triplon bound state configuration should be a stripe configuration and they conjecture that this has been seen numerically. The state then should be clarified - perhaps a figure would be useful here. For example does Ref [20] contain the same state that the authors are describing? "devised" should perhaps be "explored".

4 In Appendix B it would be useful to have a figure illustrating, for one case, the kinds of virtual processes that enter into the calculation. Why are spin operators mixed between upper and lower case in (12), (18)? In the main text, it is stated that Heff^(2) contains 4, 5, 6 particle terms presumably because the second order terms can couple up to three J bonds through two J' bonds. A reason like this would be good to mention explicitly. And what does "negligibly small" mean for these terms? Perhaps add a sentence in Appendix B to clarify which couplings are plotted in Figure 6 (for example which of the K1,2,3 are plotted?). Also, Figure 6 should have Roman font for vertical axes and the caption should clarify the horizontal axis (presumably J'_xy=J'_z).

5 Conclusions: the accomplishments and speculations would be better separated in the conclusions: for example the speculation about the zigzag ordering at finite temperature, the possible significance of the 2/15 states at the phase boundary and the stripe ordering of two-triplon bound states could be moved to a final paragraph as interesting points to explore further.

6 References: It might be worth mentioning a few points in connection to SrCu2(BO3)2: (i) the presence of weak Dzyaloshinskii-Moriya terms (T. Room, D. Huvonen, U. Nagel, J. Hwang, T. Timusk, H. Kageyama, Phys. Rev. B 70, 144417 (2004); F. Levy, I. Sheikin, C. Berthier, M. Horvatic, M. Takigawa, H. Kageyama, T. Waki and Y. Ueda, EuroPhys. Lett. 81, 67004 (2007); J. Romhanyi, K. Totsuka, and K. Penc, Phys. Rev. B 83, 024413 (2011); Tianqi Chen, Chu Guo, Pinaki Sengupta, and Dario Poletti Phys. Rev. B 101, 064417 (2020)) (ii) the topological triplon work of Judit Romhanyi, Karlo Penc & R. Ganesh, Nature Communications volume 6, Article number: 6805 (2015) (iii) perhaps also mention recent interest in the plaquette/Neel transition sparked by Ref [28]. (iv) Presence of low energy bound states: [13-16, 20] already referenced; add also [22].

  • validity: good
  • significance: ok
  • originality: good
  • clarity: ok
  • formatting: good
  • grammar: good

Author:  Taras Verkholyak  on 2021-10-05  [id 1809]

(in reply to Report 4 on 2021-06-07)
Category:
answer to question
reply to objection
correction
validation or rederivation
pointer to related literature

We are grateful to the referee for the high value of our manuscript and the insightful comments and suggestions.

Response to the requested changes:

Referee says:
1 The phase diagram in figure 8 is the central result but the abundance of detail obscures the paper's message. So, it would be good to open the paper with two schematic phase diagrams for fix J'/J and varying field: one for Ising-Heisenberg model, one for the effective Hamiltonian as a guide to the reader. In the latter schematic, please spell out that the hopping terms are not treated exactly and perhaps include the critical fields notation in (8). Since SrCu2(BO3)2 is frequently mentioned, the authors may consider including a similar rough field phase diagram for this material. Equations or at least detailed description of how to obtain the phase boundaries should be referenced from the caption.

Response:
We added the phase diagram requested by the referee and discussed it in the introduction.

Referee says:
2 If I understand correctly the 1/8 state in Fig. 7 has degenerate row and column shifts and Fig 7(b) has the middle row shifted from (a). Maybe highlight this feature in the caption. Clarify the sentence "On the other hand, the correlated hopping...": in what way is the correlated hopping suppressed? Relatedly, the fractional plateau phases are stated to be exact eigenstates of the full effective Hamiltonian so it should be possible to quote the energies including H_t, but these couplings are absent in eq. (8). Do they drop out somehow or are they actually not included?

Response:
In fact, the 1/8 plateau appeared to be even more degenerate (see new Appendix C). We added the corresponding comment to Sec.3 (page 9) and to the caption of Fig.7 (Fig.8 in the revised manuscript).

The correlated tunneling 't' is absent in Eqs. (7), (8) because the action of H_t on the states corresponding to 1/4 and 1/3 plateaus is zero. It can be easily verified, since any move of a triplon on the initial configurations violates the hard-core condition, and therefore is projected out.

Referee says:
3 Section 4 could be much clearer. The fact that this is an approximate treatment should be clarified. The precise form of the phase boundaries that go into Fig 8 should be spelt out. The restricted mobility of the quasiparticles is a crucial point but it is not evident to me from the Fig 9. nor the surrounding discussion how this constraint arises. The projectors (9) could perhaps be relegated to an appendix. In Eq. (10) presumably the P projectors should have labels. "wave" is confusing terminology - perhaps change to just "state" or "pi state". "At this critical field...should evolve": can this sentence be removed? And "evolve" in the conclusions should perhaps be "arise"? The authors seem to be saying that the two-triplon bound state configuration should be a stripe configuration and they conjecture that this has been seen numerically. The state then should be clarified - perhaps a figure would be useful here. For example does Ref [20] contain the same state that the authors are describing? "devised" should perhaps be "explored".

Response:
We modified Sec.4 for clarity. Now we start from finding the eigenstate and the eigenenergy of a single bound triplon state, and identified the critical field when the state of bound triplons appears. This is a rigorous result for the effective Hamiltonian (6). The second part of Sec.4 regards the restricted model of bound triplons as hard-core bosons where we discuss the possible origin of the quantum phase of bound triplons.

We also modified Fig.9 (Fig.11 in the revised version) to get better understanding how the hard-core condition for a bound triplon is obtained. We do leave the term free-wave of bound triplon, since it reflects the delocalized feature of these excitations.

We removed the statement about similarity of the stripe phases with Ref.20 ([22] in the revised version), since it is not evident that they are related to each other.

Referee says:
4 In Appendix B it would be useful to have a figure illustrating, for one case, the kinds of virtual processes that enter into the calculation. Why are spin operators mixed between upper and lower case in (12), (18)? In the main text, it is stated that Heff^(2) contains 4, 5, 6 particle terms presumably because the second order terms can couple up to three J bonds through two J' bonds. A reason like this would be good to mention explicitly. And what does "negligibly small" mean for these terms? Perhaps add a sentence in Appendix B to clarify which couplings are plotted in Figure 6 (for example which of the K1,2,3 are plotted?). Also, Figure 6 should have Roman font for vertical axes and the caption should clarify the horizontal axis (presumably J'_xy=J'_z).

Response:
We added Fig. 12 illustrating virtual processes for the correlated hopping terms and the corresponding text after Eq. (32).

The capital 'S' means the total spin on a dimer. We used the both notations for the spins and total spin on a dimer for the sake of compactness. We already introduced its notation after Eq.(3). But for the consistency now we added this definition also after Eq.(12).

Regarding the four-, five, six-particle terms, we envisage that each next term is about one order of magnitude smaller. At least it can be observed that three-particle couplings are of about order of magnitude smaller than the pair ones (see Fig. 7). Negligibly small means that it has no effect on the ground-state phase diagram at low densities. But we agree that it was not completely appropriate statement, therefore, we changed the text (see page 9).

The relations between the coupling constant K's introduced in Appendix B, and parameters used in the effective Hamiltonian in the main text is given at the end of Appendix B after Eq. (37).

Referee says:
5 Conclusions: the accomplishments and speculations would be better separated in the conclusions: for example the speculation about the zigzag ordering at finite temperature, the possible significance of the 2/15 states at the phase boundary and the stripe ordering of two-triplon bound states could be moved to a final paragraph as interesting points to explore further.

Response:
We changed the conclusion to emphasize the main findings of the present work which are now clearly separated from the conjectures put forward by the present calculation.

Referee says:
6 References: It might be worth mentioning a few points in connection to SrCu2(BO3)2: (i) the presence of weak Dzyaloshinskii-Moriya terms (T. Room, D. Huvonen, U. Nagel, J. Hwang, T. Timusk, H. Kageyama, Phys. Rev. B 70, 144417 (2004); F. Levy, I. Sheikin, C. Berthier, M. Horvatic, M. Takigawa, H. Kageyama, T. Waki and Y. Ueda, EuroPhys. Lett. 81, 67004 (2007); J. Romhanyi, K. Totsuka, and K. Penc, Phys. Rev. B 83, 024413 (2011); Tianqi Chen, Chu Guo, Pinaki Sengupta, and Dario Poletti Phys. Rev. B 101, 064417 (2020)) (ii) the topological triplon work of Judit Romhanyi, Karlo Penc & R. Ganesh, Nature Communications volume 6, Article number: 6805 (2015) (iii) perhaps also mention recent interest in the plaquette/Neel transition sparked by Ref [28]. (iv) Presence of low energy bound states: [13-16, 20] already referenced; add also [22].

Response:
We added the missed references and the discussion about the Dzyaloshinskii-Moriya interaction.

---

## Round 4 · Referee Report · Anonymous (Referee 3) · 2021-11-17

Report

The authors' minimal replies and alterations do nothing substantive to address the multiple issues described in detail in the previous 2 rounds of refereeing. The manuscript remains unsuitable for publication.

---

## Round 4 · Referee Report · Anonymous (Referee 5) · 2021-11-17

Report

In my opinion, the manuscript has been improved sufficiently to warrant
publication. The issues I had raised previously have been resolved.
Of course, the presentation can always be even clearer, but this would
require a total re-shaping of the article. Eventually, the authors are responsible
for their text.

---

## Round 4 · Referee Report · Anonymous (Referee 4) · 2021-11-18

Report

I am happy with the answers to my questions and the corresponding changes to the paper.

---

## Round 4 · Author Response

Dear Editor,

We are grateful to the Referees for the careful reading and valuable remarks and suggestions. We have taken into account and addressed most of the remarks in the revised version of the manuscript, whereas we provided the rebuttal against a few critical comments. We also submitted our responses to the Referee reports and the list of changes.

We herewith submit a revised version of our manuscript which we hope is suitable for the publication in SciPost Physics.

Yours sincerely,
Taras Verkholyak, Jozef Strečka

---

## Round 4 · List of Changes

page 6:
We changed J' to J'_{xy} in Eq.(5).

page 7:
The text after Eq.(5) is modified.

page 8:
We added the discussion regarding the correlated hopping terms obtained within previously considered perturbative treatments.

page 9:
Figure 6: we added the definition for the arrows to the figure caption.

page 12:
We changed the last sentence of Sec. 3 due to the remark of referee 1.

pages 15-16:
The text after Eq. (14) has been extended.

page 16:
Fig. 11 has been modified to show the possible configuration for the stripe-like phase of bound triplons.
The caption of Fig.11 has been extended.

---

## Editorial Decision

published